



# The CALIPSO Version 4 Automated Aerosol Classification and Lidar Ratio Selection Algorithm

Man-Hae Kim[1], Ali H. Omar[2], Jason L. Tackett[3], Mark A. Vaughan[2], David M. Winker[2], Charles R. Trepte[2], Yongxiang Hu[2], Zhaoyan Liu[3], Lamont R. Poole[3], Michael C. Pitts[2], Jayanta Kar[3], Brian E. Magill[3]

[1]NASA Postdoctoral Program (USRA), Hampton, VA, USA
[2]NASA Langley Research Center, Hampton, VA, USA
[3]Science Systems and Applications, Inc., Hampton, VA, USA

*Correspondence to*: Man-Hae Kim (man-hae.kim@nasa.gov)

**Abstract.** The Cloud-Aerosol Lidar with Orthogonal Polarization (CALIOP) version 4.10 (V4) level 2 aerosol data products, released in November 2016, include substantial improvements to the aerosol subtyping and lidar ratio selection algorithms. These improvements are described along with resulting changes in aerosol optical depth (AOD). The most fundamental change in V4 level 2 aerosol products is a new algorithm to identify aerosol subtypes in the stratosphere. Four aerosol subtypes are introduced for the stratospheric aerosols: polar stratospheric aerosol (PSA), volcanic ash, sulfate/other, and smoke. The tropospheric aerosol subtyping algorithm was also improved by adding the following enhancements: (1) all aerosol subtypes are now allowed over polar regions, whereas the version 3 (V3) algorithm allowed only clean continental and polluted continental aerosols; (2) a new "dusty marine" aerosol subtype is introduced, representing mixtures of dust and marine aerosols near the ocean surface; and (3) the "polluted continental" and "smoke" subtypes have been renamed "polluted continental/smoke" and "elevated smoke", respectively. V4 also revises the lidar ratios for clean marine, dust, clean continental, and elevated smoke subtypes. As a consequence of the V4 updates, the mean 532 nm AOD retrieved by CALIOP has increased by 0.044 (0.036) or 52 % (40 %) for nighttime (daytime). Lidar ratio revisions are the most influential factor for AOD changes from V3 to V4, especially for cloud-free skies. Preliminary validation studies show that the AOD discrepancies between CALIOP and AERONET/MODIS (ocean) are reduced in V4 compared to V3.

## 1 Introduction

The Cloud-Aerosol Lidar with Orthogonal Polarization (CALIOP) flown aboard the Cloud-Aerosol Lidar and Infrared Pathfinder Satellite Observations (CALIPSO) platform has been providing unique vertical profile measurements of the Earth's atmosphere on a global scale since June 2006 (Winker et al., 2010). Data products derived from the CALIOP measurements are distributed worldwide from the Atmospheric Sciences Data Center (ASDC) located at the National Aeronautics and Space Administration (NASA) Langley Research Center (LaRC). In addition to detailed spatial and optical properties of detected layers, CALIOP also provides essential information on layer types for both clouds and aerosols.

Currently, CALIOP is the only space-based sensor that observes and reports the vertical distributions of aerosol spatial and optical properties over the globe, and thus CALIOP data products offer substantial research advantages in aerosol studies. For example, CALIOP aerosol data have been widely used to evaluate aerosol model simulations for several aerosol types (e.g., Yu et al., 2010; Ford and Heald, 2012; Koffi et al., 2012; Nabat et al., 2013; Nowottnick et al., 2015), and to investigate spatio-temporal distribution



and transport of several major aerosol types, such as dust and smoke aerosols (e.g., Mona et al., 2012; Guo et al., 2017; Senghor et al., 2017; Marinou et al., 2017; Wu et al., 2017). Aerosol subtyping is important all by itself for identifying aerosol by type. But it is also important for the CALIOP level 2 retrievals of aerosol optical properties. The aerosol lidar ratio, a key parameter for the extinction retrieval, is determined for each aerosol subtype based on measurements, modeling, and the cluster analysis of a

multiyear Aerosol Robotic Network (AERONET) dataset (Omar et al., 2005; Omar et al., 2009). Because the lidar ratio is one of the largest sources of uncertainty in the CALIOP aerosol optical depth (AOD, the CALIOP aerosol classification and lidar ratio selection algorithm plays a critical role in the aerosol extinction retrieval and resulting AOD (Young et al., 2013).

In version 3 (V3) and earlier, the CALIOP level 2 aerosol classification and lidar ratio selection algorithm defined six aerosol
types: clean marine, dust, polluted continental, clean continental, polluted dust, and smoke (Omar et al., 2009). Each type is assigned an extinction-to-backscatter ratio (i.e., lidar ratio) with an associated uncertainty that defines the limits of its expected natural variability. Since the V3 release, several limitations of the V3 aerosol subtyping algorithm have come to light. For instance, mixtures of dust and marine aerosol were frequently classified as polluted dust (Burton et al., 2013), which is intended to be a mixture of dust and smoke or urban pollution. In polar regions, Asian dust and smoke from boreal fires were forced to be classified
as either clean continental or polluted continental, as the only aerosol subtypes allowed over snow, ice, or tundra. The algorithm for identifying smoke also caused some layers at the bases of elevated smoke plumes to be misclassified as clean marine. Finally, all features detected above the tropopause were generically classified as "stratospheric features" and were not given aerosol subtypes, thereby missing an opportunity to identify volcanic aerosol in the stratosphere.

The conclusions from numerous studies assert that the AOD reported in the CALIOP V3 data products typically underestimates coincident AOD measurements and/or retrievals acquired using various space-borne, airborne, and ground-based instruments (e.g., Redemann et al., 2012; Schuster et al., 2012; Kim et al., 2013; Omar et al., 2013; Rogers et al., 2014). Additional CALIOP analyses using opaque water clouds as a constraint in the retrieval (Hu, 2007) show similar results (Liu et al. 2015). However, the Moderate Resolution Imaging Spectroradiometer (MODIS) AOD retrievals (collection 5) are subject to several sources of error which mostly
tend to produce high biases in AOD (Kittaka et al., 2011). Campbell et al. (2012) compare with the US Navy Aerosol Analysis and Prediction System (NAAPS), which assimilates a quality-screened version of MODIS AOD, and find that the CALIOP AOD is consistent with NAAPS over ocean and somewhat higher over land.

There are two primary sources for the CALIOP AOD differences relative to other measurements and retrievals: aerosol layer
detection failures and inaccurate lidar ratios. Rogers et al. (2014) compared CALIOP AOD with NASA LaRC airborne High Spectral Resolution Lidar (HSRL) and found that the undetected aerosols in the free troposphere introduce a mean underestimate of 0.02 in the CALIOP column AOD in the data set examined. Kim et al. (2017) retrieved aerosol extinction for the undetected aerosol layers and found a global mean undetected layer AOD of 0.031. Toth et al. (2018) reported that 45% of daytime cloud-free V3 level 2 aerosol profiles have no aerosol detected within the profile (AOD = 0). They found the mean collocated MODIS and
AERONET AODs at 550 nm are near 0.06 and 0.08, respectively, for the CALIOP profiles without aerosols. Several other studies also suggest that the weakly backscattering aerosols which are undetected by CALIOP's layer detection algorithm can contribute to a low CALIOP AOD estimate relative to other sensors (Kacenelenbogen et al., 2011; 2014; Thorsen et al., 2017). Whereas layer detection failure always contributes to low bias, misclassification of aerosol subtypes and inaccurate aerosol lidar ratios can result in both high and low biases in CALIOP AOD. Burton et al. (2013) compared CALIOP V3 aerosol subtype product with NASA



LaRC airborne HSRL measurements. They compared 109 underflights of the CALIOP orbit track and found that 80 % of the CALIOP desert dust layers, 62 % of the marine layers and 54 % of the polluted continental layers agreed with HSRL classification results. However, the agreement was less for smoke (13 %) and polluted dust (35 %) layers. Recent studies suggest that the lidar ratios assigned by the V3 CALIOP aerosol classification and lidar ratio selection algorithm are at least partially responsible for

biases in the CALIOP V3 AOD for clean marine (Bréon, 2013; Rogers et al., 2014; Dawson et al., 2015) and dust aerosols (Burton et al., 2012; Schuster et al., 2012; Amiridis et al., 2013; Nisantzi et al., 2015; Liu et al., 2015).

The CALIOP version 4.10 (V4) level 2 aerosol data products, released in November 2016, contain substantial updates to aerosol type classification and to aerosol lidar ratio assignments, made in response to many of the results reported in the studies described

above. This paper introduces the V4 updates in the CALIOP level 2 aerosol subtyping algorithms and changes to the characteristic lidar ratios for different aerosol subtypes. The resulting AOD differences between V3 and V4 are investigated by categorizing the factors that can contribute to the AOD changes. Lastly, we compare CALIOP AOD with AERONET and MODIS for both versions as an initial validation of the CALIOP V4 AOD.

## 2 Algorithm updates for CALIOP version 4 aerosol level 2 products

The CALIOP V4 level 2 data products contain substantial refinements over V3 and earlier releases (Liu et al., 2018; Avery et al., 2018; Young et al., 2018). The most fundamental changes in V4 level 2 aerosol products are the introduction of a new 'dusty marine' aerosol subtype in the troposphere, and the addition of new aerosol subtypes to classify aerosol layers newly identified in the stratosphere. Because the cloud aerosol discrimination (CAD) algorithm is now applied to all layers detected (Liu et al., 2018), those features that were previously classified as generic "stratospheric" layers in V3 and earlier are now identified as either clouds

or aerosols. Consequently, the V4 level 2 aerosol subtyping algorithm now distinguishes between tropospheric and stratospheric aerosols. An entirely new algorithm has been implemented to identify aerosol subtypes in the stratosphere, and the algorithm for identifying tropospheric aerosol types has been substantially updated. The changes made to the tropospheric algorithm are described in detail first, followed by details on the new stratospheric aerosol subtyping algorithm.

### 2.1 Aerosol subtypes in the troposphere

The CALIOP V3 aerosol classification algorithm uses altitude, location, surface type, estimated particulate depolarization ratio ($\delta_p^{est}$), and integrated attenuated backscatter ($\gamma'$) to identify the aerosol subtype (Omar et al., 2009). Figure 1 shows the decision tree used to determine the V3 and V4 tropospheric aerosol subtypes. The major updates implemented in the V4 tropospheric aerosol subtyping algorithm include: introducing the dusty marine aerosol subtype (by adding the red shaded part in Fig. 1), allowing all aerosol subtypes over polar regions (by removing blue shaded part in Fig. 1), and revising the operational definitions for the

polluted continental and smoke aerosol types.



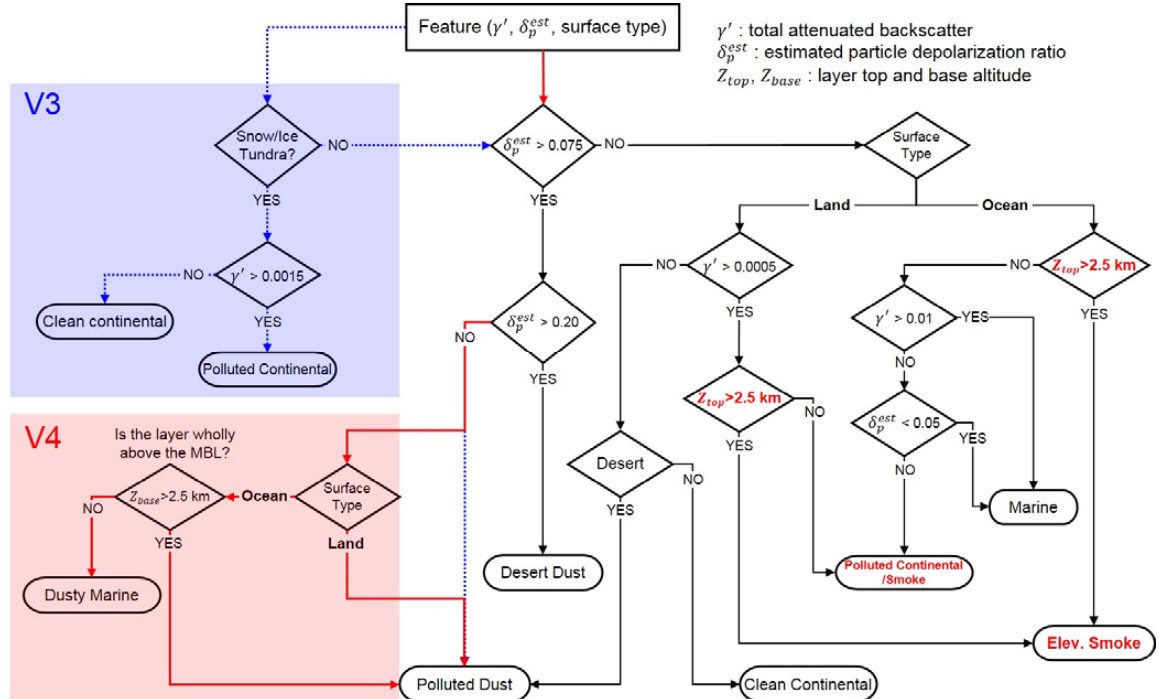

**Figure 1: Flowchart of the CALIPSO aerosol subtype selection scheme for tropospheric aerosols. Blue-shaded part and blue-dotted arrows are used in V3 but removed in V4. Red shaded parts with thick red arrows are newly added in V4. The nomenclatures for "polluted continental" and "smoke" are revised to "polluted continental/smoke" and "elevated smoke" V4. The definition for "elevated" is revised in V4 to mean layers with tops higher than 2.5 km above ground level.**

### 2.1.1 A new aerosol subtype: dusty marine

In V4, a new "dusty marine" aerosol type is introduced to identify mixtures of dust and marine aerosol and thus account for the frequent occurrence of mixtures of dust and marine aerosols that are misclassified as polluted dust over global oceans in V3. Dusty marine occurs most frequently when Saharan dust is transported across the Atlantic Ocean and settles into the marine boundary layer (MBL) as it approaches North and Central America ( Liu et al., 2008; Groß et al., 2016; Kuciauskas et al., 2018). In V3, many of these layers are misclassified as polluted dust, an aerosol type intended to represent mixtures of dust + smoke and dust + polluted continental aerosols. In both V3 and V4, polluted dust is assigned a lidar ratio of 55 ± 22 sr. Using data acquired during CALIPSO validation flights over the Caribbean Sea, Burton et al. (2013) compared CALIOP V3 aerosol classifications with measurements made by the NASA LaRC airborne HSRL on the NASA B200 aircraft. For those layers that CALIOP V3 classified as polluted dust, the HSRL measured a median lidar ratio 35 sr, thus strongly suggesting that these aerosols were a combination of dust + marine aerosol, and not the combination of dust + smoke modeled by the CALIOP polluted dust type. As shown in Fig. 2(a), 40 % to 50 % of aerosol samples over the Caribbean in JJA at night are classified as polluted dust in V3. During the daytime in V3 (Fig. 2(b)), polluted dust accounts for 10 % to 30 % of aerosol samples identified over remote oceanic regions (e.g., the South Pacific Ocean) where the occurrence of mixtures of dust and smoke is less probable.





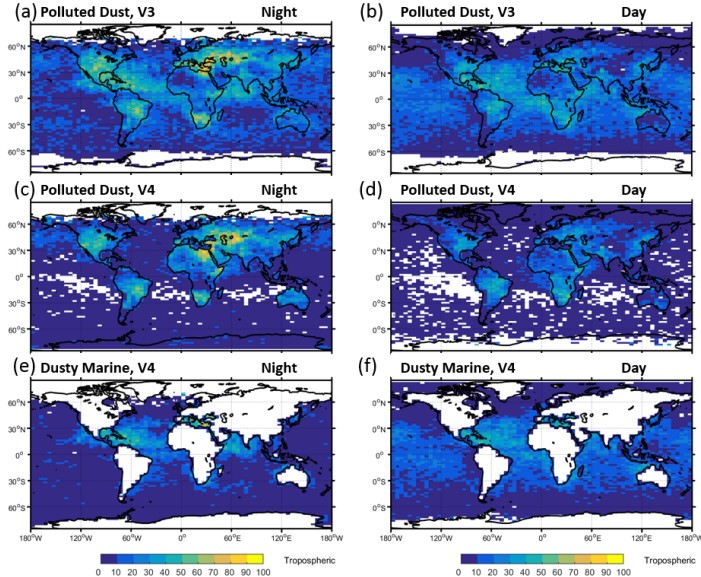

**Figure 2: Frequency of occurrence of aerosol samples classified as polluted dust in V3 at night and day (a, b), polluted dust in V4 at night and day (c, d) and dusty marine in V4 at night and day (e, f). June-August 2007.**

5 The polluted dust classification occurred in V3 because these layers are mildly depolarizing, having estimated particulate depolarization ratios ($\delta_p^{est}$) between 0.075 and 0.20 (Omar et al., 2009). The estimated particulate depolarization ratio is layer-integrated volume depolarization ratio which is corrected to account for the molecular contribution, defined as

$$\delta_p^{est} = \frac{\delta_v'[(R_{mas}-1)(1+\delta_m)+1]-\delta_m}{(R_{mas}-1)(1+\delta_m)+\delta_m-\delta_v'}, \tag{1}$$

where $\delta_v'$ is the layer-integrated volume depolarization ratio, $R_{mas}$ is the mean attenuated scattering ratio and $\delta_m$ is the molecular depolarization ratio. Here, $\delta_v'$ is defined as

$$\delta_v' = \frac{\sum_{z_{base}}^{z_{top}}[\beta_\perp'(z)]}{\sum_{z_{base}}^{z_{top}}[\beta_\parallel'(z)]}, \tag{2}$$

where z is altitude and the subscripts 'top' and 'base' refer to the top and base of the detected aerosol layer.

When a dust layer, having $\delta_p^{est} > 0.20$, mixes with non-depolarizing marine aerosol, the layer-averaged $\delta_p^{est}$ decreases below 0.20 and the aerosol is classified as polluted dust in V3. This explains the enhanced frequency of V3 polluted dust classifications over
20 the Caribbean in JJA (Fig. 2(a)). In other oceanic regions where dust + marine or dust + smoke mixtures are less probable (again, the remote South Pacific Ocean), the frequency of polluted dust is overestimated in V3 for at least two reasons. First, $\delta_p^{est}$ is a noisy quantity that has a positively skewed distribution that is exacerbated by solar background noise during the daytime. Additionally, occasional high biases can arise from residual single-shot resolution cloud contamination within the MBL. This




makes the 0.075 lower $\delta_p^{est}$ threshold easier to exceed in these situations. Second, $\delta_p^{est}$ is overestimated in V3 because attenuation from overlying layers was not accounted for in the $\delta_p^{est}$ computation (Burton et al., 2013) in V3. This oversight has been corrected in V4.

To identify dust + marine aerosol mixtures in V4, dusty marine layers are defined as moderately depolarizing ($0.075 < \delta_p^{est} < 0.20$) aerosol layers over ocean having base altitudes below 2.5 km above mean sea level (an upper limit for the MBL; Winning et al., 2017). After implementation of this new aerosol subtype, the frequency of oceanic layers classified as polluted dust decreased substantially. Given that the CALIOP polluted dust subtype is explicitly modeled as a mixture of dust and smoke (Omar et al., 2009), the V4 spatial distributions of polluted dust over the Caribbean and remote Pacific Ocean shown in Figs. 2c and 2d present
a more likely scenario than the V3 distributions shown in Figs 2a and 2b. 30-50 % of aerosol samples over the Caribbean are classified as dusty marine in JJA as shown in Figs. 2(e-f). Note that the dusty marine frequency is enhanced over remote oceanic regions during the daytime (Fig. 2(f)). This is due to the noisiness of $\delta_p^{est}$ in daytime. The AODdifferences from misclassifying pure marine (lidar ratio of 23 sr) as dusty marine (37 sr) are substantially less than they would otherwise be if these same layers were instead misclassified as polluted dust (55 sr).

### 2.1.2 Aerosol subtypes in polar regions

As indicated in Fig. 1, the V3 aerosol classification algorithm allows only clean continental and polluted continental subtypes over the polar regions when the surface type is either snow, ice, or tundra. This was based on the assumption that the aerosol found in the polar regions – particularly in the Arctic in winter – are pollutants from industrialized areas transported poleward (Stohl, 2006; Stone et al., 2008). During the spring phase of the ARCTAS campaign (Jacob et al., 2010), however, the poleward transport of
multiple plumes of Asian dust and smoke from boreal fires were observed, highlighting the importance of these other aerosol types. The contribution of smoke to the aerosol found in the Arctic – primarily from boreal forest fires and high latitude agricultural fires – is now well documented (e.g., Stohl et al., 2007; Warneke et al., 2010; Di Pierro et al., 2011; Markowicz et al., 2016). Records in polar ice and snow cores show that dust has been transported to the Arctic and Antarctic since geologic times (e.g., Lunt and Valdes, 2001; Fischer et al., 2007). While there are dust sources at high latitudes in both the northern (Alaska, Canada, Greenland,
and Iceland) and southern (Antarctica, New Zealand, and Patagonia) hemispheres (Bullard et al., 2016), they are minor sources and the primary source of dust transported to the Arctic is the Asian deserts. Huang et al. (2015) investigate a large-scale dust storm that occurred in East Asia using ground-based and space-borne remote sensing measurements, NCEP/NCAR reanalysis data and a HYSPLIT trajectory analysis. They found that the dust storm was rapidly transported to the Arctic from its source region within 5 days.

Because of the recent realization of the importance of smoke and dust over the Arctic, the V4 aerosol classification algorithm no longer uses snow, ice, and tundra as decision points, but instead uses uniform aerosol typing criteria for the entire Earth (Fig. 1). As a consequence, all CALIOP aerosol subtypes may now be identified in polar regions. Figure 3 shows the dust plume reported by Huang et al. (2015). The plume is well captured by CALIOP, as shown in total attenuated backscatter (Fig. 3(a)) and
depolarization ratio (Fig. 3(b)). However, the V3 algorithm identifies the plume as dust/polluted dust at latitudes less than 56°N but as polluted continental/clean continental above 56°N (Fig. 3(c)). The sea surface changes from open water to ice at this point, and hence the aerosol subtyping is forced by the V3 polar region loop (Fig. 1). In V4, this plume is correctly classified as dust (Fig. 3(d)).



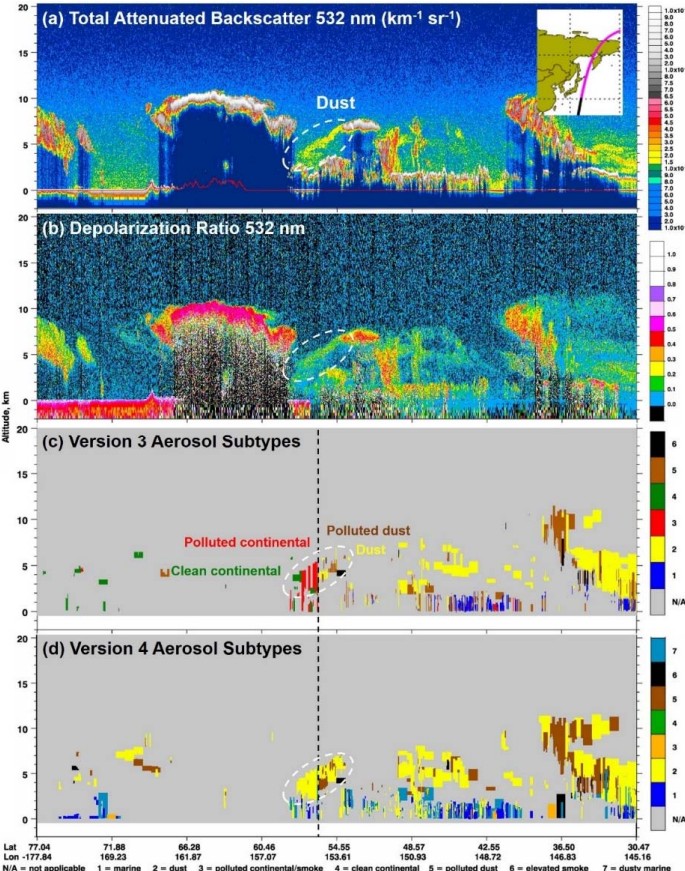

**Figure 3. CALIOP measurements of (a) 532 nm total attenuated backscatter, (b) 532 nm depolarization ratio, and aerosol subtypes in (c) V3 and (d) V4 on March 22, 2010 between 16:11-16:25 UTC. White dashed ellipse shows dust plume and black dashed line represents the boundary for "snow/ice, tundra" used for polar regions in V3.**

### 2.1.3 Revised aerosol subtype: elevated smoke and polluted continental/smoke

The interpretation and nomenclature of layers identified in V3 as smoke and polluted continental has been revised in V4. As in previous versions, elevated non-depolarizing aerosols are assumed to be smoke that has been injected above the planetary boundary layer (PBL). The definition for "elevated" is revised in V4 to mean layers with tops higher than 2.5 km above ground level (i.e., a simple approximation of a region above the PBL). For clarity, the name of the smoke aerosol subtype is changed to "elevated smoke" to emphasize that these layers are identified as smoke because they are elevated above the PBL. Within the PBL, the optical properties measured by CALIOP (depolarization and color ratio) are practically identical for the smoke and polluted continental subtypes, making them indistinguishable. To acknowledge the optical similarity of polluted continental and smoke, the name of this aerosol type is changed in V4 to "polluted continental/smoke". The V4 lidar ratios used in the CALIOP retrieval algorithm are identical for polluted continental/smoke and elevated smoke (70 sr at 532 nm and 30 sr at 1064 nm).



### 2.2 Stratospheric aerosols

In V4, the CAD algorithm is applied at all altitudes, including in the stratosphere. By contrast, previous versions only applied the CAD algorithm below the tropopause, classifying layers detected above the tropopause as "stratospheric features" rather than as clouds or aerosols. As a consequence, aerosol existing above the tropopause was not identified explicitly as aerosol. However, it

is well documented that certain aerosol types exist in the stratosphere. Volcanic eruptions inject ash and sulfate to high altitudes (e.g., Vernier et al., 2011; Bourassa et al., 2012). Smoke due to intense combustion or from pyro-cumulonimbus events can also breach the tropopause (e.g., Fromm et al., 2005; 2010; Trentmann et al., 2006). In the polar winter, polar stratospheric clouds (PSCs) form and the PSC composed of supercooled ternary solution (STS) is an aerosol (Pitts et al., 2009). In V4, features identified by the CAD algorithm as aerosol having 532 nm attenuated backscatter centroids (Garnier et al., 2015) above the tropopause from

the Modern-Era Retrospective analysis for Research and Applications, Version 2 (MERRA-2) reanalysis data (Gelaro et al., 2017) outside of the polar regions are classified as "stratospheric aerosols". Distinguishing among the different types of stratospheric aerosol relies primarily on latitude, temperature, and the measured properties of each layer: $\gamma'$ and $\delta_p^{est}$ at 532 nm and integrated attenuated total color ratio, $\chi'$, where $\chi'$ is the layer mean attenuated backscatter at 1064 nm divided by the layer mean attenuated backscatter at 532 nm (Liu et al., 2018).

V4 identifies four stratospheric aerosol subtypes: volcanic ash, sulfate/other, elevated smoke, and polar stratospheric aerosol (PSA). Volcanic ash is defined as an aspherical volcanic aerosol that depolarizes the 532 nm backscatter, whereas sulfate/other is defined primarily as a non-depolarizing volcanic aerosol. The "other" component of this aerosol type is the catch-all for stratospheric aerosol layers that are either weakly scattering or cannot be classified as any other type within the stratospheric

aerosol algorithm. Weakly scattering layers are not evaluated by the stratospheric aerosol subtyping algorithm because the noisy values of $\delta_p^{est}$ and $\chi'$ at low signal levels inhibit robust classifications with the threshold-based technique employed. The PSA subtype is introduced in V4 to assign a reasonable aerosol type for features detected in the polar regions during polar winter, and subsequently classified as aerosol by the CAD algorithm due to their low values of $\chi'$ and $\gamma'$. Comparison with the CALIOP L2 PSC mask product shows that PSAs are spatially correlated with the STS PSC composition class (Pitts et al., 2009; Liu et al.,

2018). However, layers assigned the PSA subtype should be interpreted carefully. For in-depth studies, the CALIPSO team recommends using the CALIOP L2 PSC mask product for analyses related to PSC composition since it is a more specialized product (Pitts et al., 2009).



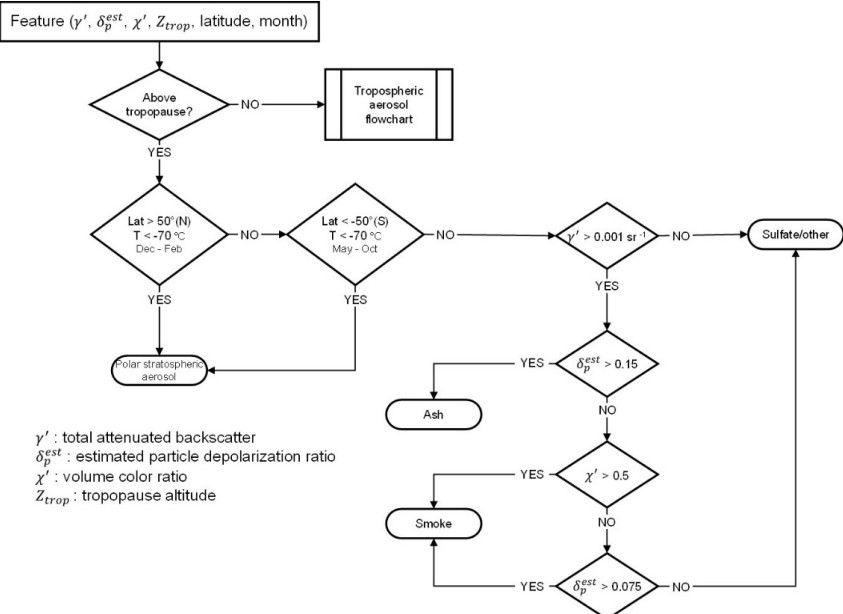

**Figure 4: Flowchart of the CALIPSO aerosol subtype selection scheme for stratospheric aerosols.**

The stratospheric aerosol subtyping algorithm is summarized by the flowchart in Fig. 4. PSAs are identified first by determining

5  if the stratospheric aerosol layer's latitude, season, and temperature at its 532 nm attenuated backscatter centroid altitude are consistent with that of PSCs. A centroid altitude temperature threshold of –70° C is implemented based on the occurrence frequency distribution of aerosol mid-layer temperatures during the Antarctic PSC season in 2008 (Fig. 5).The –70° C temperature threshold captures the increased aerosol occurrence frequency at colder temperatures, corresponding to PSC misclassified as aerosol (e.g., temperatures of less than –75° C corresponds to temperatures consistent with PSC formation in this region (Rosen et al., 1997)).

10  Latitude and seasonal constraints are applied to ensure the PSA type is assigned where and when PSC formation occurs. PSA classification is only allowed for latitudes poleward of 50° S or N, and PSC seasons for the Arctic and Antarctic regions are assumed to be December – February and May – October, respectively (Poole and Pitts, 1994).





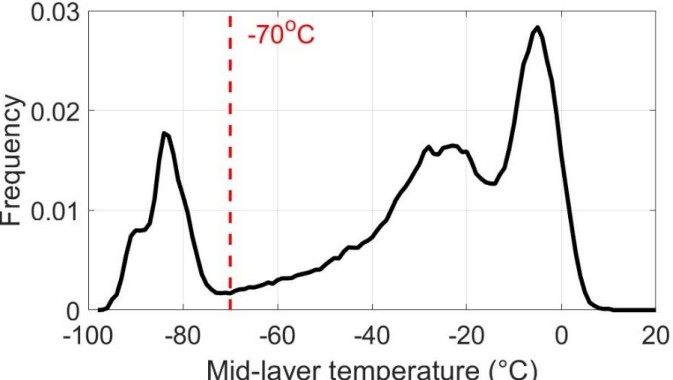

**Figure 5: Occurrence frequencyf of mid-layer temperatures for stratospheric aerosol layers south of 50°S during Antarctic PSC season, June – September 2008 at night.**

Next, in order to discriminate between volcanic ash, sulfate, and elevated smoke, the stratospheric aerosol typing algorithm evaluates layer-averaged $\delta_p^{est}$ and $\chi'$ against empirically derived thresholds. These thresholds were derived from frequency distribution analysis of $\delta_p^{est}$ and $\chi'$ measurements obtained from a manually identified subset of volcanic ash, sulfate, and high-altitude smoke layers (Fig. 6). The number of unique layers detected by CALIOP, geophysical events, and dominant aerosol types

contributing to this subset are summarized in Table 1. Note that, as previously mentioned, weakly scattering stratospheric aerosol layers (layers with $\gamma' < 0.001$ sr$^{-1}$) are directly classified as sulfate/other due to their low signal-to-noise ratio (SNR). As shown in Figs. 6(a-c), volcanic ash and volcanic sulfate are fairly well separated with respect to $\delta_p^{est}$, with ash typically having $\delta_p^{est} > 0.15$ and sulfate with $0.075 < \delta_p^{est} < 0.15$. Smoke layers are less depolarizing than volcanic ash (Fig. 6(d)), with $\delta_p^{est} < 0.15$. However, smoke layers can either be non-depolarizing or moderately depolarizing (Figs. 6(e-f)). An example of a moderately depolarizing

smoke event is the February 2009 "Black Saturday" Australian bush fire where $\delta_p^{est}$ exceeds 0.10 for many layers detected by CALIOP. Non-depolarizing smoke layers ($\delta_p^{est} < 0.075$) typically have $\chi' > 0.5$, whereas $\chi'$ is more frequently lower for moderately depolarizing smoke layers ($0.075 < \delta_p^{est} < 0.15$). Based on this analysis, the stratospheric aerosol typing algorithm depicted in Fig. 4 was constructed using the thresholds indicated by the red lines in Fig. 6.

**Table 1: Number of layers detected by CALIOP used to determine V4 stratospheric aerosol typing thresholds. Dominant aerosol type for volcanic events determined according to references in the table.**

| N Layers | Geophysical Event | Dominant Aerosol Type |
|---|---|---|
| 2274 | Puyehue-Cordón Caulle eruption, June 2011 | Volcanic ash (Bignami et al., 2014) |
| 69 | Okmok eruption, July 2008 | Volcanic ash (Prata et al., 2010) |
| 58 | Chaiten eruption, May 2008 | Volcanic ash (Prata et al., 2010) |
| 2439 | Kasatochi eruption, August 2008 | Volcanic sulfate (Krotkov et al., 2010) |
| 256 | Nabro, June 2011 | Volcanic sulfate (Theys et al., 2013) |
| 813 | Siberian fires, May-June 2012 | Smoke |
| 399 | Canadian fires, July-August 2007 | Smoke |
| 1624 | Australian bush fire, February 2009 | Smoke, depolarizing (de Laat et al., 2012) |
| 161 | Canadians fire, May 2007 | Smoke, depolarizing |





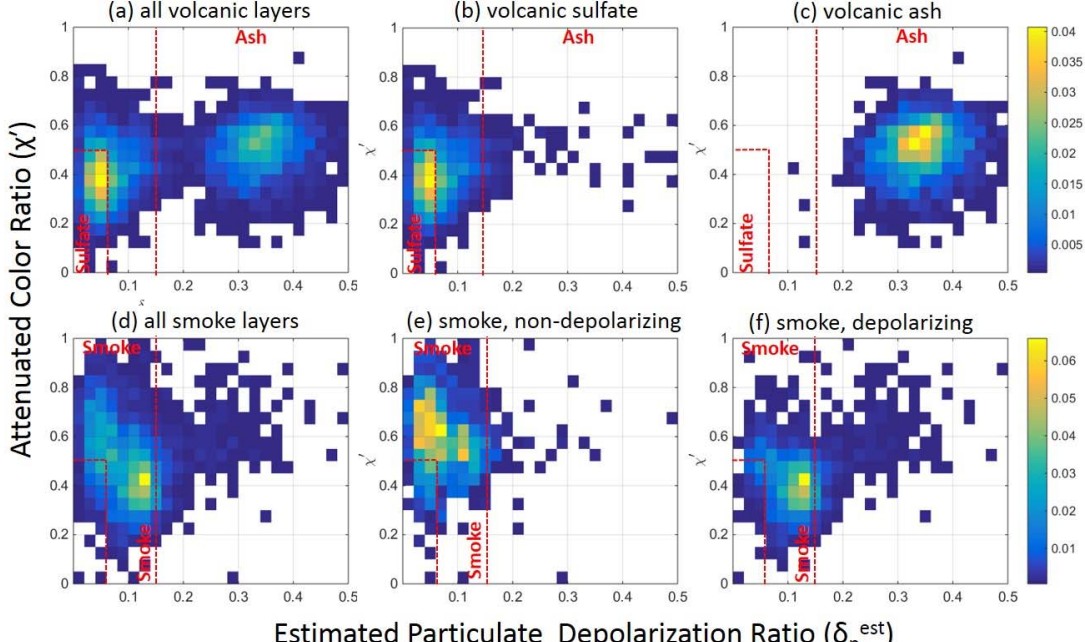

**Figure 6: Two-dimensional frequency distributions of attenuated total color ratio and estimated particulate depolarization ratio for layers in Table 1. Distributions are normalized independently by the sum of samples in the subset: (a) all volcanic layers, (b) volcanic sulfate, (c) volcanic ash, (d) all smoke layers, (e) non-depolarizing smoke, and (f) depolarizing smoke. Only layers having integrated**
**attenuated backscatter > 0.001 sr⁻¹ contribute. Red dashed lines denote the V4 stratospheric aerosol typing thresholds.**

The following examples demonstrate the strengths and limitations of the stratospheric aerosol typing algorithm. Volcanic ash is well-separated from the other types in terms of $\delta_p^{est}$ and $\chi'$, which often leads to robust subtype classifications. Figure 7 shows a scene where the algorithm correctly classifies the bulk of a volcanic ash plume from the Puyehue-Cordón Caulle eruption in June

2011. Though the northern-most layers near 42°-45° S are automatically classified as sulfate/other due to their low $\gamma'$, these layers are optically thin and the more optically thick layers of the ash plume are classified correctly. Also note that portions of the ash plume having backscatter centroids below the tropopause are misclassified as dust. This occurs because aerosol layers below the tropopause are assigned tropospheric aerosol subtypes which do not include ash. Because the noise-broadened distributions of $\delta_p^{est}$ for ash and dust measured by CALIOP have very similar characteristics, we know of no robust way to discriminate the two within

the troposphere (Winker et al., 2012). However, the lidar ratio assigned for ash is identical to that for dust (Table 2) so potential misclassifications will have minimal impact on the extinction products.

Figure 8 presents a scene where the bulk of the Nabro volcano plume from June 2011 (Theys et al., 2013) is correctly classified as sulfate. In this example, most of the layer has $\gamma' > 0.001$ and low depolarization, yielding a sulfate classification for the more

optically-thick segments. However, aa small number of layers within the plume are misclassified as smoke. This is expected because of the overlap in the frequency distributions of $\delta_p^{est}$ and $\chi'$ for sulfate (Fig. 6(b)) and smoke (Fig. 6(d)). The optical properties used for these two types are not as well separated from each other as they are from volcanic ash, so some misclassifications can occur. Additionally, volcanic sulfate within the troposphere will be assigned a tropospheric aerosol type, usually elevated smoke or clean continental if weakly scattering. The last example in Fig. 9 shows an observation of a depolarizing




smoke plume from the Black Saturday Australian bush fire in February 2009 (Pumphrey et al., 2011; de Laat et al., 2012). The majority of the plume above the tropopause is correctly classified as smoke, with the minority misclassified as ash due to $\delta'$ exceeding 0.15. The remainder of the plume below the tropopause is misclassified as dust and polluted dust, again due to elevated depolarization.. In all of these examples, the V3 data products classified the layers detected above the tropopause as a generic

5 'stratospheric layer' without applying any further subtyping. In V4, these same layers are most often correctly classified as aerosols by the new CAD algorithm (Liu et al., 2018). Similarly, the new stratospheric aerosol subtyping algorithm is largely successful in identifying the correct aerosol subtype.

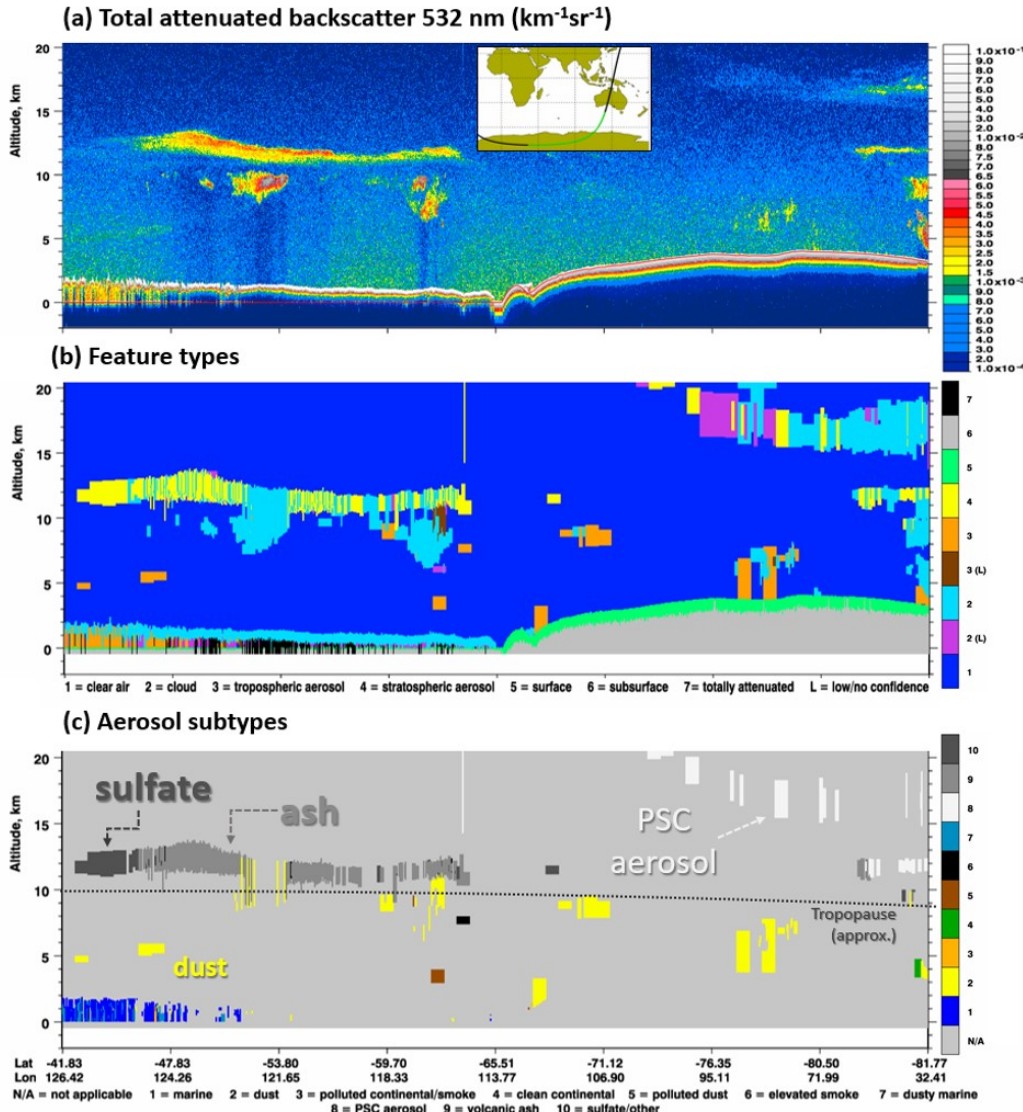

10 **Figure 7: CALIOP observations of the Puyehue-Cordón Caulle volcano plume on June 20, 2011 between 16:50-17:00 UTC. (a) Total attenuated backscatter 532 nm, (b) V4 feature type classification, and (c) V4 aerosol subtypes, where the dashed line indicates the approximate location of the tropopause. The satellite ground track is indicated by the green section on inset map in panel (a).**





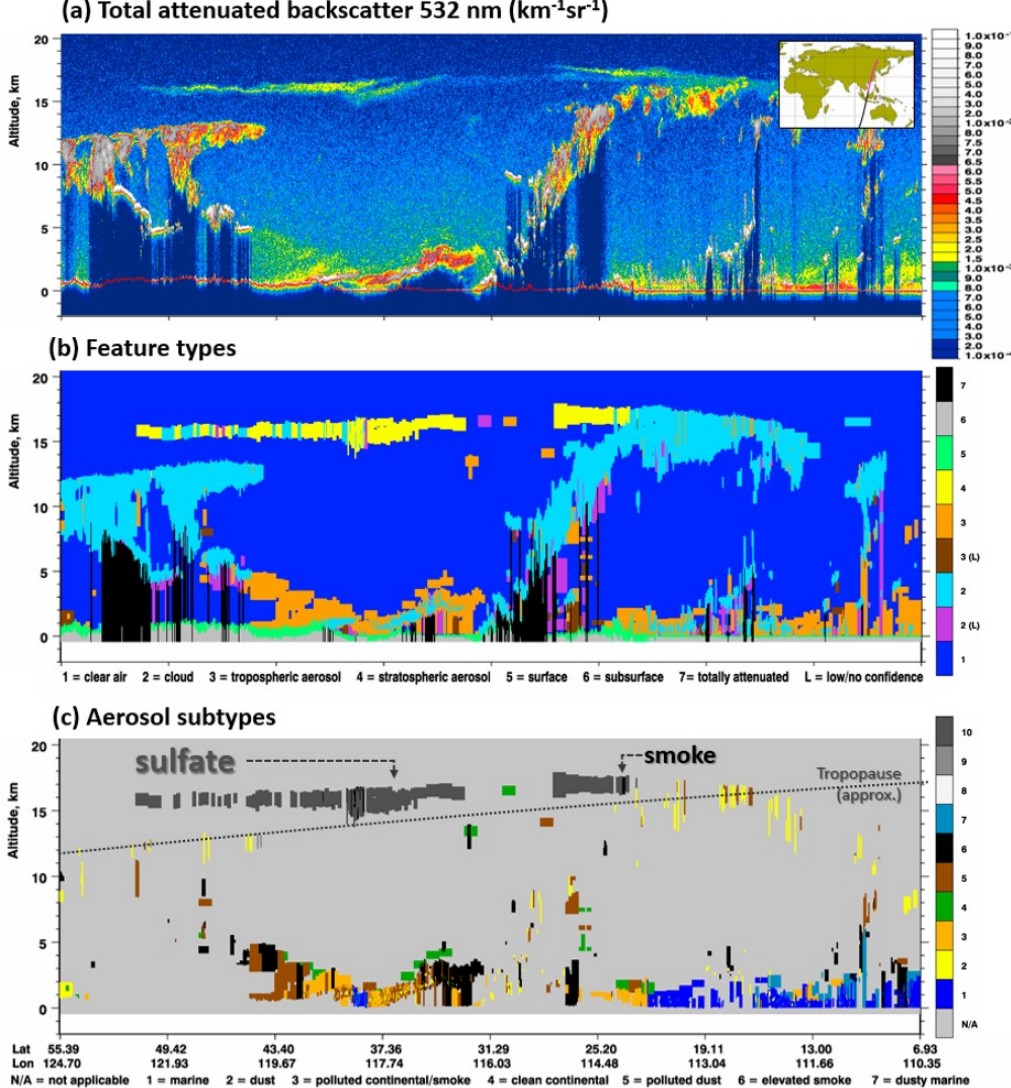

**Figure 8: CALIOP observations of the Nabro volcano plume on June 18, 2011 between 18:13-18:26 UTC. (a) Total attenuated backscatter 532 nm, (b) V4 feature type classifications, and (c) V4 aerosol subtypes; dashed line indicates approximate location of the tropopause. The satellite ground track is indicated by the magenta section on inset map in panel (a).**





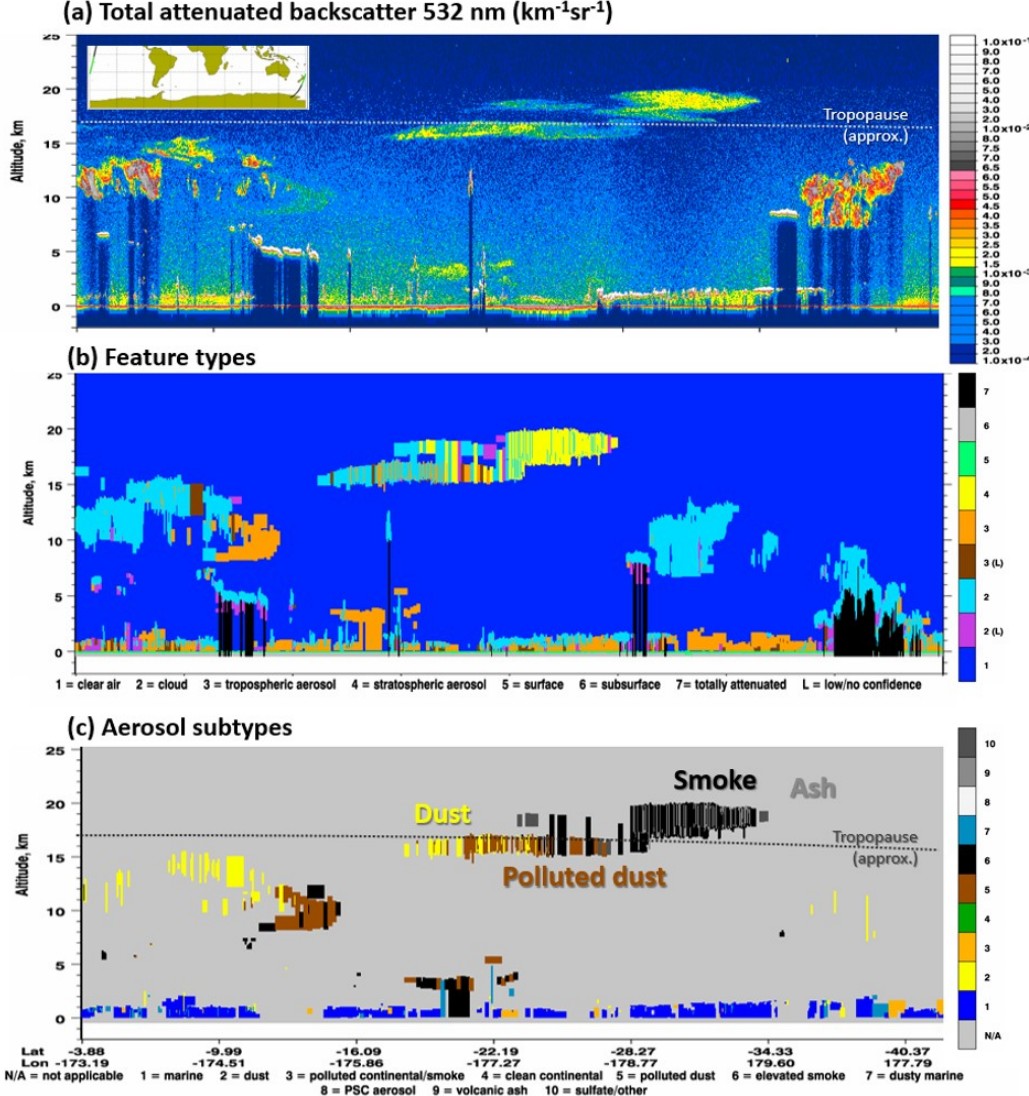

**Figure 9: CALIOP observations of smoke plumes from the Australian bush fire on February 15, 2012 between 13:19-13:32 UTC. (a) Total attenuated backscatter 532 nm, (b) V4 feature type classification, and (c) V4 aerosol subtypes, where the dashed line indicates the approximate location of the tropopause. The satellite ground track is indicated by green section on inset map in panel (a).**

### 2.3 Subtype Coalescence Algorithm for AeRosol Fringes (SCAARF)

In previous data releasesreleases, "fringes" at the bases of dense aerosol plumes were at times misclassified as an aerosol subtype inconsistent with the parent plume. These fringes typically lie below rapidly attenuating aerosol layers, and are detected at 20 km or 80 km horizontal resolution (Vaughan et al., 2009). An example is shown in Figs. 10(a-b), where fringes at the base of an

elevated smoke plume are misclassified as clean marine aerosol. In this case, the fringes are misclassified because the layers are non-depolarizing and have top altitudes just below the 2.5 km altitude threshold that would have otherwise caused them to be





correctly classified as elevated smoke according to the revised definition of "elevated" in V4 (Sect. 2.1.3). Given that this is an elevated plume not in contact with any aerosol beneath, it is reasonable to expect that the misclassified fringes at the base of the plume have the same aerosol subtype as the adjacent smoke layers. This same argument can be made for other aerosol types that contiguously span large horizontal distances (e.g., dust plumes, marine aerosol, volcanic ash, and volcanic sulfate).

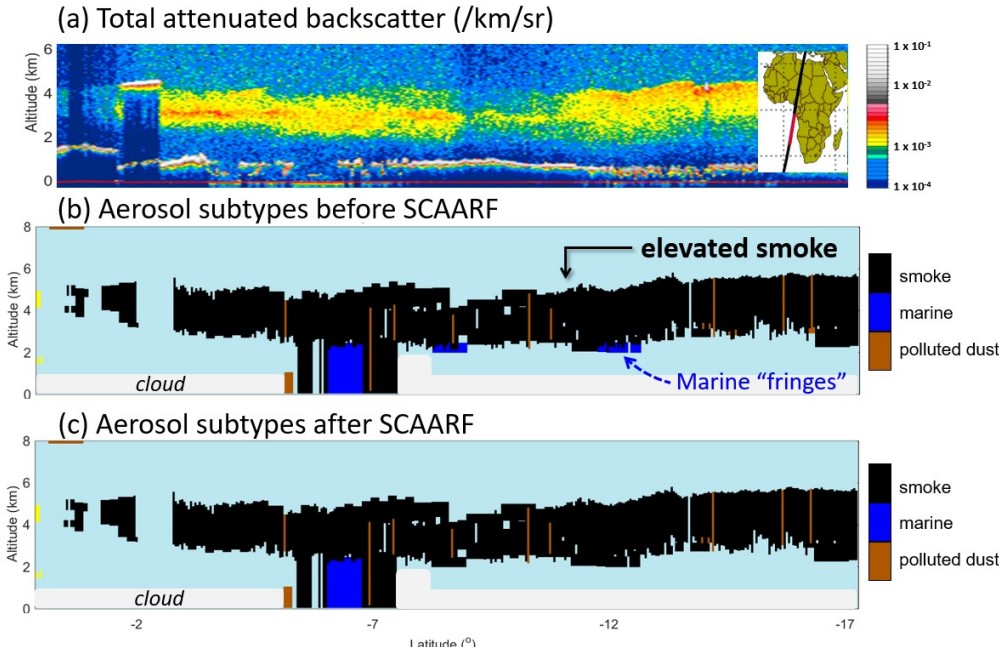

**Figure 10: CALIOP observations of a smoke plume off the west coast of Africa on September 14, 2008 from approximately 1:08 to 1:10 UTC. (a) Total attenuated backscatter 532 nm and aerosol subtype classification (b) before and (c) after SCAARF is implemented. Inset map in panel (a) shows CALIOP ground track shown in red. Aerosol subtypes: elevated smoke (black), clean marine (blue), polluted dust (brown).**

In general, aerosol layers detected by CALIOP that are in contact with other aerosol layers are assumed to be of the same aerosol type. For this reason, V4 implements the Subtype Coalescence Algorithm for AeRosol Fringes (SCAARF), which re-classifies the aerosol subtype of these lower fringes to match the dominant subtype of the adjacent overlying layers. According to SCAARF,

fringes are defined as aerosol layers detected at 20 km or 80 km horizontal resolutions that are vertically adjacent to the base(s) of aerosol layers detected at finer spatial resolution (i.e., they are adjacent to more strongly scattering features). At least 50 % of the horizontal extent of the fringe candidate must be in contact with aerosol overhead. SCAARF is applied to all tropospheric and stratospheric aerosol layers meeting this "fringe" criteria. The dominant adjacent aerosol subtype is determined from the number of 5 km resolution samples vertically adjacent to the fringe. When two adjacent aerosol subtypes exists with equal frequency (i.e.,

neither is dominant in terms of number), the fringe is changed to match the subtype of the adjacent layers that are most similar to the fringe in terms of $\delta_p^{est}$ and $\chi'$. This is the subtype with the minimum Euclidian distance $r_i$ between $\bar{\delta}_p^{est}$ and $\bar{\chi}'$ of the parent and $\bar{\delta}_p^{est}$ and $\bar{\chi}'$ of the fringe;; i.e.,

$$r_i = \sqrt{\left(\delta_{p,fringe}^{est} - \bar{\delta}_{p,subtype\,i}^{est}\right)^2 + \left(\chi'_{fringe} - \bar{\chi}'_{subtype\,i}\right)^2}, \quad \text{for } i \in [1,2].$$   (3)





Here, $\bar{\delta}^{est}_{p,subtype\,i}$ and $\bar{\chi}'_{subtype\,i}$ are the average values of $\delta^{est}_p$ and $\chi'$ for all adjacent layers having unique subtypes $i \in [1, 2]$. The fringe is changed to match subtype $i$ where $r_i = minimum(r_1, r_2)$. If three or more unique subtypes are adjacent to the fringe with equal frequency, SCAARF is not applied.

In effect, SCAARF aids in vertically homogenizing aerosol subtype classification along plume bases. Figure 10(c) demonstrates that the fringes misclassified as clean marine have been correctly classified as elevated smoke after SCAARF is implemented. Similar improvement occurs for volcanic ash layers straddling the tropopause. Lower fringes of these plumes below the tropopause would otherwise be misclassified as dust (Sect. 2.2), yet SCAARF helps retain the volcanic ash classification.

### 2.4 Aerosol lidar ratios in version 4

Table 2 shows the lidar ratios that characterized the V3 aerosol types, and the revised values used in V4 for tropospheric aerosols and the newly introduced stratospheric aerosols. Except for polluted continental and polluted dust, V4 aerosol lidar ratios have been updated to reflect the improved knowledge from measurements reported in recent literature. In addition, lidar ratios have

15 been defined for the new aerosol types: dusty marine and the stratospheric aerosol types. The modifications and new lidar ratios are based on the latest available measurements, both from CALIPSO and from other researchers and field measurements campaigns.

**Table 2: Aerosol lidar ratios with expected uncertainties for tropospheric and stratospheric aerosol subtypes at 532 nm and 1064 nm in**
20 **CALIOP version 3 and 4 aerosol retrieval algorithm.**

| Aerosol subtype | $S_{532}$ (sr) | | $S_{1064}$ (sr) | |
|---|---|---|---|---|
| | **Tropospheric aerosols** | | | |
| | **V3** | **V4** | **V3** | **V4** |
| Clean marine | 20 ± 6 | 23 ± 5 | 45 ± 23 | 23 ± 5 |
| Dust | 40 ± 20 | 44 ± 9 | 55 ± 17 | 44 ± 13 |
| Polluted continental / smoke | 70 ± 25 | 70 ± 25 | 30 ± 14 | 30 ± 14 |
| Clean continental | 35 ± 16 | 53 ± 24 | 30 ± 17 | 30 ± 17 |
| Polluted dust | 55 ± 22 | 55 ± 22 | 48 ± 24 | 48 ± 24 |
| Elevated smoke | 70 ± 28 | 70 ± 16 | 40 ± 24 | 30 ± 18 |
| Dusty marine | - | 37 ± 15 | - | 37 ± 15 |
| | **V4 Stratospheric aerosols** | | | |
| Polar stratospheric aerosol | 50 ± 20 | | 25 ± 10 | |
| Volcanic ash | 44 ± 9 | | 44 ± 13 | |
| Sulfate / other | 50 ± 18 | | 30 ± 14 | |
| Smoke | 70 ± 16 | | 30 ± 18 | |

### 2.4.1 Clean marine

The lidar ratio and uncertainties for clean marine aerosol are modified from 20 ± 6 sr in V3 to 23 ± 5 sr at 532 nm in V4. This change is consistent with results reported from numerous field campaigns since the launch of CALIPSO. Papagiannopoulos et al.

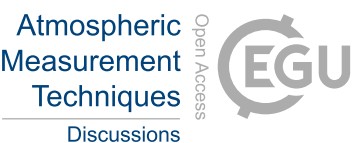

(2016) report lidar ratio of 23 ± 3 sr for marine aerosol from the European Aerosol Research Lidar Network (EARLINET), based mainly on multi-wavelength Raman lidar systems. Müller et al. (2007) also show that mean lidar ratio for marine aerosols in the PBL is 23 ± 3 sr from the Second Aerosol Characterization Experiment (ACE 2) and 23 ± 5 sr from the Indian Ocean Experiment (INDOEX). In an analysis of collocated HSRL measurements acquired during dedicated CALIPSO underflights, Rogers et al.

(2014) find that the median lidar ratio for layers identified as marine aerosol is 23 sr.

With respect to marine lidar ratios at 1064 nm, Josset et al. (2012) applied the Synergized Optical Depth of Aerosols (SODA) technique (Josset et al., 2011) to CALIOP measurements at both 532 nm and 1064 nm, and found no spectral dependence in the retrieved lidar ratios. Similarly, Sayer et al. (2012) calculate lidar ratios for marine aerosols from AERONET island sites, spread

throughout the world's oceans, and find little spectral dependence. Based on these studies, the CALIOP lidar ratio for clean marine at 1064 nm is changed from 45 ± 23 sr in previous versions to 23 ± 5 sr, which is same value used at 532 nm in V4.

### 2.4.2 Dust

There are multiple studies reporting dust lidar ratios larger than 40 ± 20 sr at 532 nm, the value used in previous versions of

CALIOP algorithm (e.g., Liu et al., 2002; Müller et al., 2007; Wandinger et al., 2010; Schuster et al., 2012; Papagiannopoulos et al., 2016). Liu et al. (2015) derive dust lidar ratio directly from the CALIOP measurements using a constrained opaque water cloud technique (Hu, 2007) and find mean/median lidar ratios of 45.1/44.4 ± 8.8 sr for moderately dense Sahara dust layers. Furthermore, from measurements of ground-based Raman lidars and airborne HSRL, no wavelength dependence of dust lidar ratio is found among 355, 532, and 1064 nm (Tesche et al., 2009). Based on these studies, dust lidar ratios in V4 are modified to 44 ± 9 sr at 532

nm and 44 ± 13 sr at 1064 nm. Though lidar ratios for dust show regional variability ranging mostly from 35 to 60 sr (Cattrall et al., 2005; Schuster et al., 2012; Mamouri et al., 2013; Nisantzi et al., 2015), only a single value is used in the V4 algorithm. Implementing a regionally varying lidar ratio for dust is complicated due to uncertainties in determining the dust source regions for transported dust, and introducing unnatural discontinuities in global dust AOD. The uncertainty in V4 dust lidar ratio of 20 % (30 %) at 532 nm (1064 nm) accounts for the regional variability.

### 2.4.3 Polluted continental and elevated smoke

Polluted continental lidar ratios at 532 nm and 1064 nm are unchanged from V3 to V4, at 70 ± 25 sr and 30 ± 14 sr respectively. For elevated smoke, the 532 nm lidar ratio is the same in both versions, but, based on a study by Liu et al. (2015), the uncertainty is reduced from 70 ± 28 sr in V3 to 70 ± 16 sr in V4. The lidar ratio at 1064 nm for elevated smoke is changed from 40 ± 24 sr in

V3 to 30 ± 14 sr in V4 (Sayer et al., 2014), so that the V4 value for smoke now matches that of polluted continental. Elevated smoke detected in the stratosphere also uses these same lidar ratios.

### 2.4.4 Clean continental

In V3, the lidar ratios used for the clean continental subtype were 35 ± 16 sr at 532 nm and 30 ± 17 sr at 1064 nm. The 532 nm

value is generally consistent with the background aerosol lidar ratios being reported in the literature available at the start of the mission (e.g., Voss et al., 2001; Ansmann et al., 2001). However, in an extensive CALIPSO validation study, Rogers et al. (2014)



found that the mean 532 nm lidar ratio measured by the LaRC HSRL in layers identified by CALIOP as clean continental was 53 ± 11 sr. In V4, the 532 nm lidar ratio for clean continental aerosols is therefore changed to 53 ± 11 sr.  Because the LaRC HSRL only measures lidar ratio at 532 nm, no additional information on lidar ratios at 1064 nm is available from Rogers et al. (2014). Consequently, the V4 lidar ratio for clean continental aerosol at 1064 nm remains unchanged from V3.

### 2.4.5 Polluted dust and dusty marine

Validation with MODIS and airborne HSRL measurements show that CALIOP V3 AODs and lidar ratios appear to be biased high for layers in some regions which are classified as polluted dust (Kim et al., 2013; Burton et al., 2013; Rogers et al., 2014). This is especially true over the ocean. Since CALIOP V3 did not account for mixtures of dust and sea salt which are frequent in the MBL,

the bias is likely a result of dust + marine misclassified as dust + smoke. For this reason, the lidar ratios for polluted dust remains same in V4 as in V3, and a new aerosol subtype, dusty marine, is introduced to reflect the correct mixture. The lidar ratios for polluted dust are unchanged from their V3 values, at 55 ± 22 sr at 532 nm and 48 ± 24 sr at 1064 nm. Based on an assumed external mixture of dust and marine aerosol (65:35 by surface area), the lidar ratios for dusty marine are 37 ± 15 sr at both wavelengths using mean lidar ratios of 44 sr and 23 sr for pure dust and clean marine, respectively. Using the NASA HSRL in the MBL in the

Caribbean region, Rogers et al. (2014) found lidar ratios of 37 ± 11 sr for mixtures of dust and marine aerosols. As is the case for both dust and marine, the lidar ratios for dusty marine combination are spectrally independent. However, the lidar ratio uncertainties ascribed to the dusty marine type are larger than either dust or marine alone. The range of uncertainty of the dusty marine lidar ratio in V4 (15 sr) is greater than the uncertainty for these mixtures in Rogers et al. (2014) and accounts for a large range of possible surface area mixing ratios of dust and marine aerosols in the ambient MBL.

### 2.4.6 Volcanic ash

Default lidar ratios for volcanic ash are set in V4 to match that of the dust subtype: 44 ± 9 sr at both 532 nm and 1064 nm. These lidar ratios were selected following Winker et al. (2012), where it is shown that the size distribution, composition, and shape of transported volcanic ash particles are similar to desert dust. This was based on comparisons of in-situ aircraft measurements by

Schumann et al. (2011) during the April 2010 Eyjafjallajökull volcano eruption and Saharan dust properties described by Omar et al., (2010). In reality, lidar ratios for volcanic ash vary depending on the composition of the plume and circumstances of the eruption (water vapor content, mineralogy, plume age, injection height, etc.). Recent studies have found higher lidar ratios for volcanic ash (Table 3). For example, lidar ratios of 50 ± 10 sr were retrieved for volcanic ash transported over Europe during the April 2010 Eyjafjallajökull volcano eruption (Ansmann et al., 2011; Groß et al., 2012), with mean lidar ratios of 60 ± 5 sr at Leipzig

on 16 April 2010 (Ansmann et al., 2010). Recently, Prata et al. (2017) used constrained CALIOP retrievals to estimate mean particulate lidar ratios of 69 ± 13 sr for volcanic ash from the June 2011 Puyehue-Cordón Caulle eruption. The results of these studies suggest that 44 sr is likely near the lower limit of natural variability. Given the large variability in lidar ratios for volcanic ash and the evolving state of knowledge, the CALIPSO team is further studying the representativeness of this lidar ratio based on information gained since the 2016 release of the version 4 level 2 products.



**Table 3: Mean 532 nm lidar ratios reported in the literature for volcanic ash and volcanic sulfate.**

| Mean lidar ratio (532 nm) | Volcanic Eruption | Reference |
|---|---|---|
| **Ash dominant aerosol** | | |
| 50 ± 10 sr | Eyjafjallajökull, 2010 | Ansmann et al., 2011; Groß et al., 2012 |
| 60 ± 5 sr | Eyjafjallajökull, 2010 | Ansmann et al., 2010 |
| 69 ± 13 sr | Puyehue-Cordón Caulle, 2011 | Prata et al., 2017 |
| **Sulfate dominant aerosol** | | |
| 30 – 50 sr | Kasatochi, 2008; Sarychev Peak, 2009 | Mattis et al., 2010 |
| 48 sr | Nabro, 2011 | Sawamura et al., 2012 |
| 55 sr | Mt. Etna, 2002 | Pappalardo et al., 2004 |
| 55 ± 4 sr | Sarychev Peak, 2009 | O'Neill et al., 2012 |
| 63 ± 14 sr | Sarychev Peak, 2009 | Prata et al., 2017 |
| 65 ±10 sr | Kasatochi, 2008 | Hoffmann et al., 2010 |
| 66 ± 19 sr | Kasatochi, 2008 | Prata et al., 2017 |

### 2.4.7 Sulfate/other

Default lidar ratios for sulfate/other are 50 ± 18 sr and 30 ± 14 sr at 532 nm and 1064 nm, respectively. Researchers have previously reported independent lidar measurements of sulfate-rich volcanic plumes from Mt. Etna 2002, Kasatochi 2008, Sarychev Peak

2009, and Nabro 2011 eruptions, summarized in Table 3. Lidar ratios from these studies range from 30 – 66 sr, with CALIOP-constrained lidar ratio retrievals reported by Prata et al. (2017) on the high end: 63 ± 14 sr and 66 ± 19 sr for the Kasatochi and Sarychev Peak eruptions, respectively. The 532 nm lidar ratio for sulfate is consistent with these studies given the variability of measured lidar ratios and the 35 % uncertainty implemented with the default lidar ratio, yielding 50 ± 18 sr. Independent measurements of 1064 nm lidar ratios for volcanic sulfate are sparse in the literature. The default lidar ratio value of 30 sr, however,

is consistent with Jäger and Hofmann (1991) which reported measured background stratospheric aerosol levels of 35 sr and 39 sr for years 1979-1980 and 1986-1987, respectively. The 1064 nm lidar ratio is also consistent with that of the CALIOP model for polluted continental aerosol which is, in part, modelled after sulfate (Omar et al., 2009).

### 2.4.8 Polar stratospheric aerosol

Default lidar ratios for PSA are 50 ± 20 sr at 532 nm and 25 ± 10 sr at 1064 nm. As discussed in Sect. 2.3, these aerosol layers

exhibit a qualitative spatial correlation with the supercooled ternary solution (STS) composition class in the CALIPSO level 2 PSC mask product. These lidar ratios and their wavelength dependence are consistent with theoretical Mie scattering calculations for STS droplets at pressures typical of the Arctic stratosphere.

### 3 Aerosol subtyping changes from version 3 to version 4

The performance and final results delivered by the V4 aerosol subtyping algorithm are affected by V4 changes to several other

algorithms that occur earlier in the level 2 processing scheme. The CALIOP V4 level 1 data significantly improved the calibration of the CALIOP attenuated backscatter coefficients ($\beta'$) at both 532 nm and 1064 nm (Getzewich et al., 2018; Kar et al., 2018; Vaughan et al., 2018a). In particular, calibration coefficients at 532 nm decreased by ~3 % to ~12 %, depending on latitude and season, resulting in a concomitant increase in $\beta'$ at 532 nm. The increased magnitude of $\beta'$ at 532 nm subsequently yields an increase in the number of tenuous layers detected by the CALIOP feature finder. The V4 CAD algorithm features entirely new



probability distribution functions (PDFs) that are now more sensitive to the presence of lofted aerosols (Liu et al., 2018). As a consequence, the V4 data products show improvements in the identification of high altitude smoke plumes and Asian dust layers, which in earlier versions were often classified as cirrus clouds. Also, the V4 analyses use a completely new algorithm to detect the Earth's surface detection (Vaughan et al., 2018b). This new technique demonstratesan improvement over the V3 method in turbid

5 atmospheres, while maintaining equal or better performance in clear skies. As a result of this improved surface detection scheme, there are fewer opaque layers identified in V4 than there were in V3, especially at night. Because regions below layers previously classified as opaque are now scanned for the presence of atmospheric features, there is also a slight increase in the number of cloud and aerosol layers reported. Taken together, these changes yield an increase in the absolute number of layers classified as aerosols in V4 relative to V3.

The feature type changes from/to aerosol, aerosol subtypes changes, and resulting AOD changes between V3 and V4 are analyzed using the atmospheric volume description (AVD) reported in level 2 aerosol profile product. AVD reports both feature type and aerosol/cloud subtype for each 5 km x 60 m (5 km x 180 m for above 20.2 km) range bin. The feature types include clear air, cloud, tropospheric aerosol, stratospheric feature/aerosol (V3/V4, respectively), surface, subsurface, and totally attenuated regions (i.e.,

15 beneath layers classified as opaque in V3 but reclassified as transparent in V4). Table 4 shows changes in feature type and aerosol subtype between V3 and V4 using the AVD data in the level 2 profile products. Though the table contains all changes among feature types and aerosol subtypes, in this study we focus solely on changes in the distribution of aerosol subtypes and the downstream effects of these changes in the global and regional distributions of AOD.





**Table 4: Feature type and aerosol subtype changes in the CALIOP level 2 atmospheric volume description (AVD) between version 3 (V3) and version 4 (V4) from 2007 to 2009. Each (i,j) component of the Table represents what fraction (expressed as a percentage) of type i in V3 changes to type j in V4, thus the summation of each column equals to 100 (%). Since the total number of each type is different, relative total amounts for each type are shown as normalized total for both columns and rows which are normalized to total number of bins for V3 aerosol.**

| V4 \ V3 | Total Atten. | Clear | Cloud | Surface | Aerosol | CM* | Dust | PC* | CC* | PD* | Smoke | Strato. Feature | Normalized Total |
|---|---|---|---|---|---|---|---|---|---|---|---|---|---|
| Total Atten. | 84.30 | 0.03 | 1.42 | 12.98 | **0.09** | 0.13 | 0.06 | 0.06 | 0.01 | 0.11 | 0.02 | **0.00** | 2.09 |
| Clear | 7.14 | 98.97 | 1.92 | 7.81 | **6.28** | 4.43 | 4.99 | 8.88 | 14.87 | 7.10 | 8.61 | **4.11** | 34.18 |
| Cloud | 3.49 | 0.34 | 92.39 | 7.76 | **6.99** | 5.35 | 4.64 | 9.59 | 11.56 | 7.85 | 13.29 | **61.56** | 3.08 |
| Surface | 3.68 | 0.03 | 0.23 | 56.45 | **0.35** | 0.15 | 0.51 | 0.70 | 0.38 | 0.43 | 0.35 | **-** | 0.17 |
| Tropo. Aerosol | **1.40** | **0.48** | **3.76** | **15.00** | **85.22** | **89.94** | **89.78** | **80.66** | **63.86** | **84.37** | **70.90** | **0.30** | **1.18** |
| CM* | 0.35 | 0.10 | 0.50 | 10.63 | **33.78** | 80.83 | 0.40 | 21.99 | 11.60 | 6.15 | 0.12 | **-** | 0.41 |
| Dust | 0.35 | 0.09 | 1.68 | 0.90 | **17.16** | 0.19 | 73.25 | 1.74 | 5.02 | 6.63 | 0.89 | **0.10** | 0.26 |
| PC*/ Smoke | 0.08 | 0.04 | 0.18 | 0.84 | **5.66** | 0.93 | 0.48 | 33.08 | 9.69 | 5.20 | 18.13 | **-** | 0.08 |
| CC* | 0.01 | 0.02 | 0.02 | 0.11 | **0.66** | 0.00 | 0.05 | 0.58 | 8.82 | 0.54 | 0.65 | **0.02** | 0.01 |
| PD* | 0.16 | 0.11 | 0.75 | 0.66 | **10.81** | 0.02 | 10.45 | 5.45 | 11.26 | 32.36 | 6.94 | **0.07** | 0.17 |
| Elev. Smoke | 0.15 | 0.07 | 0.36 | 0.11 | **7.53** | 4.09 | 0.27 | 11.60 | 10.17 | 4.72 | 43.67 | **0.11** | 0.12 |
| DM* | 0.31 | 0.05 | 0.27 | 1.76 | **9.62** | 3.87 | 4.89 | 6.23 | 7.31 | 28.78 | 0.50 | **-** | 0.13 |
| Strato. Aerosol | **0.00** | **0.16** | **0.27** | **-** | **1.07** | **0.00** | **0.01** | **0.11** | **9.31** | **0.14** | **6.84** | **34.03** | **0.13** |
| PSA | 0.00 | 0.02 | 0.11 | - | **0.06** | - | - | 0.01 | 1.21 | 0.00 | 0.01 | **13.84** | 0.04 |
| Volcanic Ash | 0.00 | 0.00 | 0.01 | - | **0.00** | - | 0.00 | - | 0.01 | 0.00 | 0.00 | **0.25** | 0.00 |
| Sulfate/ other | 0.00 | 0.13 | 0.14 | - | **1.00** | 0.00 | 0.01 | 0.11 | 8.06 | 0.13 | 6.81 | **19.73** | 0.10 |
| Smoke | 0.00 | 0.00 | 0.01 | - | **0.01** | - | - | 0.00 | 0.04 | 0.01 | 0.02 | **0.21** | 0.00 |
| Total | 100 | 100 | 100 | 100 | **100** | 100 | 100 | 100 | 100 | 100 | 100 | **100** | |
| Normalized Total | 2.40 | 34.22 | 2.90 | 0.11 | **1.00** | 0.38 | 0.21 | 0.06 | 0.05 | 0.22 | 0.08 | **0.18** | 40.82 |

\* CM=clean marine, PC=polluted continental, CC=clean continental, PD=polluted dust, DM=dusty marine, PSA=polar stratospheric aerosol.

### 3.1 Feature type changes

Feature type changes between V3 and V4 are predominantly due to extensive changes in the calibration coefficients reported the CALIOP level 1 product, which in turn required major revisions of the probability distribution functions (PDFs) that drive the CAD algorithm (Liu et al., 2018). Changes to the surface detection algorithm (Vaughan et al., 2018b) also contribute, but to a significantly lesser extent. In order to quantify how the occurrence frequency of aerosol types has changed, Table 4 reports the percent changes in "feature type" and "aerosol subtype" between V3 and V4 for all 60 m range bins in the CALIOP level 2 profile product from 2007 to 2009. Salient statistics drawn from Table 4 are given here and in subsequent sections. Within the three year analysis period, the classification of 13.7 % of the layers identified as tropospheric aerosols in V3 has changed to totally attenuated (0.1 %), clear air (6.3 %), cloud (7.0 %), or surface (0.4 %) in V4. In spite of this reduction, however, tropospheric aerosols in V4 increase by 18 % due to newly identified aerosols from regions that were identified in V3 as totally attenuated layer, clear air, cloud, or surface. The V3 CAD algorithm did not separate aerosols from clouds for layers detected in the stratosphere; instead, it identified these layers generically as "stratospheric features". In V4, however, the CAD algorithm is applied in both the troposphere and stratosphere, and thus aerosol layers are identified and classified in the stratosphere. When including stratospheric aerosol layers, aerosols increase by 31 % in V4 compared to V3. The CALIOP V4 algorithm detects more aerosol resulting in an increase





AOD. Additionally, the improved surface detection scheme of V4 results fewer opaque layers than in V3 (Vaughan et al., 2018b). Thus, regions below layers previously classified as opaque are now scanned for the presence of atmospheric features. This leads to an increase in the number of aerosol layers reported near the surface.

### 3.2 Aerosol subtype changes

The spatial distribution and frequency of occurrence of aerosols has changed from V3 to V4 for reasons described in Sect. 3.1. Similarly, enhancements to the aerosol subtyping algorithm described in Sect. 2 are responsible for changes in the spatial distributions and occurrence frequencies of the different aerosol subtypes. The net effect of these changes is demonstrated by Fig. 11, which shows the difference in aerosol type detection frequencies for JJA 2007, day and night combined. For context, Fig. 12 shows the number of aerosol samples detected during the same time period. The frequency of clean marine aerosol is slightly

reduced in V4 except for in the oceans around Antarctica (Fig. 11(a)), with most changed layers becoming dusty marine. Table 4 shows that 3.9 % of V3 clean marine aerosol is reclassified as dusty marine in V4. The increase of clean marine aerosol in V4 over the Antarctic Ocean mainly comes from clean continental and polluted continental aerosols due to the changes in aerosol subtyping algorithm over the polar regions (Fig. 1). 4.1 % of clean marine aerosol off the southwest African coast became elevated smoke in 2007-2009 (Table 4).

The revised definition for elevated smoke (Sect. 2.1.3) and the implementation of SCAARF (Sect. 2.3) are responsible for correcting the frequency of elevated smoke classifications in this region in V4 (Fig. 11(f)). Additionally, the revised elevated smoke definition is responsible for the changes in polluted continental/smoke and elevated smoke classifications over southern Africa in Fig. 11(c) and Fig. 11(f), respectively. During JJA, smoke from biomass burning is ubiquitous in this region, so a smoke

aerosol type classification is expected most often. Because the top altitudes of smoke layers within this region are often below 2.5 km above the ground level, many layers do not meet the V4 "elevated" definition, causing an increase in the frequency of polluted continental/smoke classifications and a reduction in the frequency of elevated smoke classifications as compared to V3.





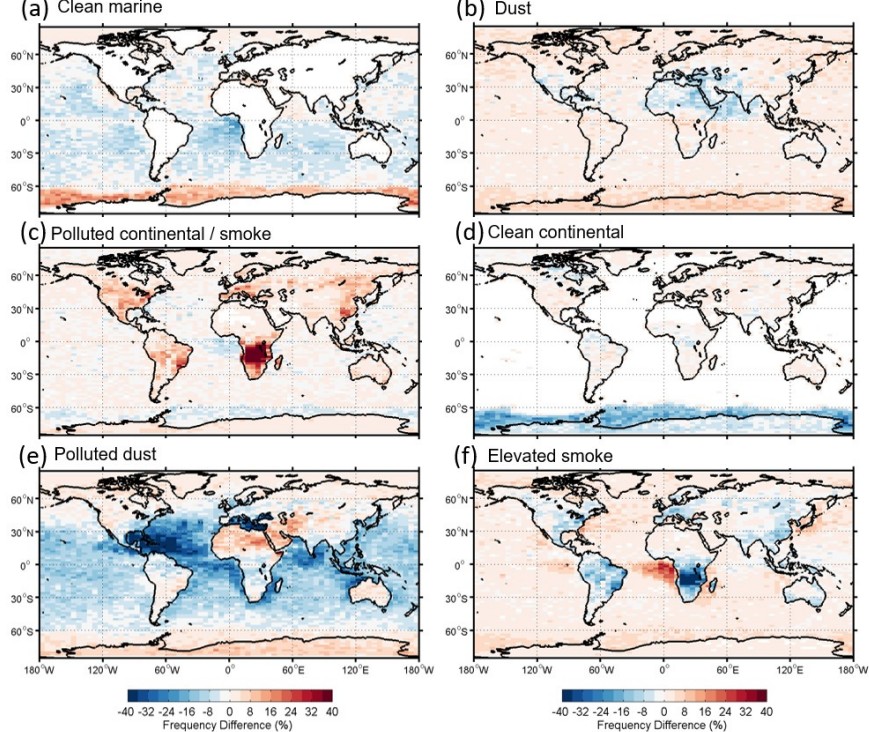

**Figure 11: Difference in frequency of occurrence of indicated aerosol type from V3 to V4 (fV4 – fV3) for aerosol subtypes common to both version, JJA 2007 day & night. Frequencies are computed from level 2 aerosol profile products as the number of aerosol samples with the indicated aerosol type divided by the total number of aerosol samples.**

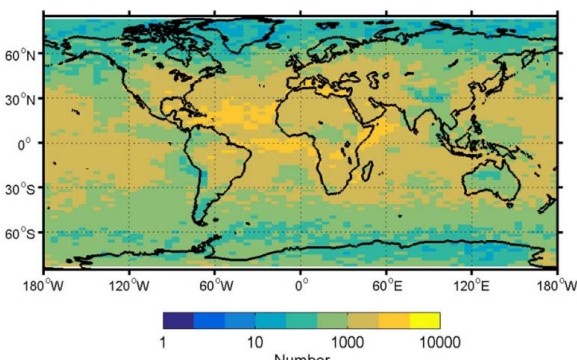

**Figure 12: Number of aerosol samples detected in V4 for JJA day & night combined, computed from level 2 aerosol profile product.**

10 As expected and intended, the introduction of the new dusty marine aerosol type (Sect. 2.1.2) has improved the aerosol subtyping over oceans in regions where mixtures of dust and urban pollution are not expected (e.g., mid-Atlantic and mid-Pacific Oceans). This is shown by a decreased frequency of polluted dust aerosol layers in V4 (Fig. 11(e)). Over the north African/Arabian dust region, the frequency of polluted dust has increased while the frequency of dust has decreased (Fig. 11(b)), in part due to correcting the overestimate of $\delta_p^{est}$ that existed in V3. This correction in $\delta_p^{est}$ also caused about 4.9 % of V3 dust to become dusty marine in





V4 (Table 4). Though the fraction of aerosol classified as dust has changed by a small amount, the number of dust layers at high altitudes has increased due to changes in CAD, which shows anan improved ability to correctly classify lofted dust layers as aerosols rather than cirrus clouds (Liu et al., 2018).

The frequency of clean continental aerosol has decreased over regions characterized by snow, ice and tundra (Fig. 11(d)) because all aerosol type classifications are allowed in V4 over these surface types (Sect. 2.1.1). Clean continental aerosols have mainly changed to clean marine (11.6 %), polluted dust (11.3 %), elevated smoke (10.2 %), and polluted continental/smoke (9.7 %). The increase in dust and polluted dust classifications over the Antarctic reflect type misclassifications of tenuous ice clouds and blowing snow. Only 8.8 % of clean continental aerosol layers are unchanged in V4 (Table 4).

As a global summary, Fig. 13 shows frequency distributions of aerosol subtypes for daytime and nighttime in V3 and V4, normalized by the total number of bins (day and night together for each version) that were classified as aerosol according to the AVD data from the level 2 aerosol profile products. More aerosol layers are detected at night for both V3 and V4 (Liu et al., 2018). This is expected since a higher SNR at night means the CALIOP layer detection algorithm detects more weakly scattering features

during nighttime (Vaughan et al., 2009). Clean continental is only rarely identified in V4. Clean continental was common in the polar regions, especially over the Antarctic in V3. Because V4 allows all aerosol types in the poles, the dominance of clean continental is significantly reduced, as shown in Fig. 11(d). The frequency for polluted dust is reduced for both day and night. WhileWhile part of this reduction is due to the layer attenuation corrections mentioned in Sect. 2.1.1, the predominant reason is because layers previously classified as polluted dust are now correctly classified as dusty marine in V4. Since the frequency of

occurrence of polluted dust aerosols is larger for daytime compared to nighttime over ocean in V3, as shown in Fig. 3(b), the change from polluted dust to dusty marine is relatively more frequent for daytime than nighttime. In fact, 59 % of the daytime dusty marine in V4 is polluted dust in V3, but only 42 % of the nighttime dusty marine is polluted dust in V3. Stratospheric features detected in V3 are classified as cloud or stratospheric aerosol in V4. Stratospheric features in V3 are changed to cloud more frequently for nighttime. Similarly, stratospheric aerosol in V4 is less at nighttime than daytime compared to stratospheric features

in V3.





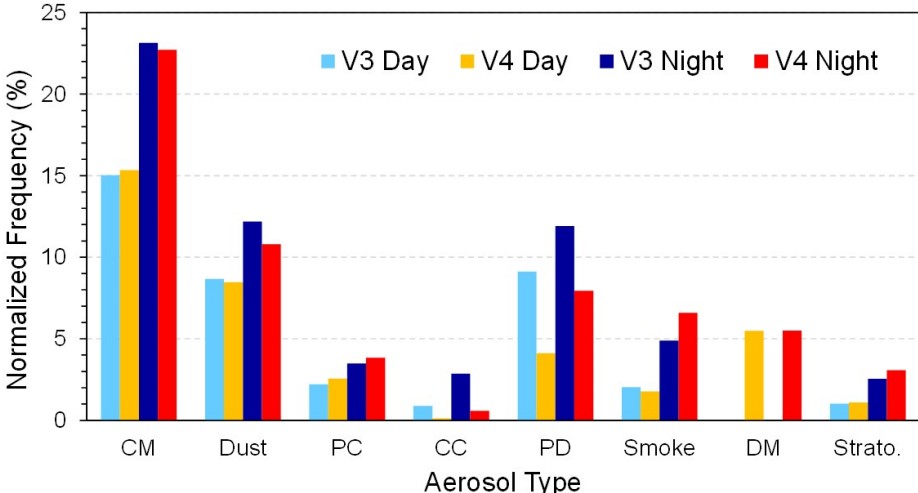

**Figure 13: Normalized frequencies of aerosol subtypes in V3 and V4 for daytime and nighttime. Note that "strato." is stratospheric feature for V3 but stratospheric aerosol for V4. CM=clean marine, PC=polluted continental, CC=clean continental, PD=polluted dust, DM=dusty marine.**

### 3.3 AOD changes

In order to compute the change in AOD from V3 to V4, CALIOP level 2 aerosol extinction profiles are vertically integrated and compared profile to profile between the two versions. Only profiles where either/both V3 or V4 contained aerosol are included in the average (in cases where only one version reports aerosol, the other version is assumed to have AOD = 0). Furthermore, only

10 aerosol bins with the extinction quality control (QC) flags of 0, 1, 16, and 18 are used (excluding 6 % of aerosol samples), which represent unconstrained retrievals which do not change the lidar ratio (0), constrained retrievals which derive an optimized lidar ratio (1), opaque layers for which the lidar ratio was not changed (16), and opaque layers where the lidar ratio was reduced to prevent the retrieval from diverging (18), respectively (Tackett et al., 2018). For 2007 to 2009, the total change in mean nighttime (daytime) CALIOP level 2 column AOD has increased from 0.084 (0.090) in V3 to 0.128 (0.126) in V4 (Table 5). Day and night

15 AOD become more comparable in V4 whereas daytime AOD is larger than nighttime in V3. Note that the mean AOD computed here is not meant to represent global conditions, but instead examines AOD changes only where AOD is detected by CALIOP.

**Table 5: Mean column AODs (± standard deviation) for CALIOP V3 and V4, computed from aerosol extinction profiles, for all-sky condition from 2007 to 2009. Profiles where either/both V3 or V4 contained aerosol layers are included in the average.**

|      | Night       | Day         |
|------|-------------|-------------|
| V3   | 0.084±0.162 | 0.090±0.150 |
| V4   | 0.128±0.242 | 0.126±0.202 |

The AOD increase from V3 to V4 is due to various factors. Using the feature types and aerosol subtypes in reported in the level 2 AVD, AOD changes attributed to layer detection, CAD, totally attenuated layers, surface detection, stratospheric aerosol classification, aerosol type, and lidar ratio are identified using the procedure diagrammed in Fig. 14. This strategy isolates changes in AOD due to each of these factors using CALIOP level 2 products from 2007 to 2009. All range bins whose feature type is




determined as aerosol by either V3, V4 or both are selected for the analyses. If a bin is identified as aerosol in one of V3 or V4 and the other is clear the corresponding AOD change is regarded as the change due to the difference of layer detection in the two versions (pathway 1 in Fig. 14). Similarly, an aerosol bin that changed from/to cloud, totally attenuated layer, surface, and stratospheric feature is counted in the AOD changes due to the updates of CAD, totally attenuated signals, surface detection, and

stratospheric aerosol in V4, respectively (pathways 2-5). When feature types in both V3 and V4 are aerosol, AOD differences can be due to aerosol subtype changes (pathway 6) or lidar ratio adjustments without changing subtype (pathway 7). If aerosol subtype is identified as polluted continental, polluted dust, or smoke in both V3 and V4, there are no changes in the aerosol subtyping (pathway 8). However, the AOD can be different between V3 and V4 even when there are "no changes" in their subtype and lidar ratio. The most likely source of these differences is changes in the magnitude of $\beta'$, either due to level 1 calibration improvements

(Kar et al., 2018; Getzewich et al., 2018) or to changes in the two-way transmittances estimated for overlying cloud and/or aerosol layers (Young et al., 2018).

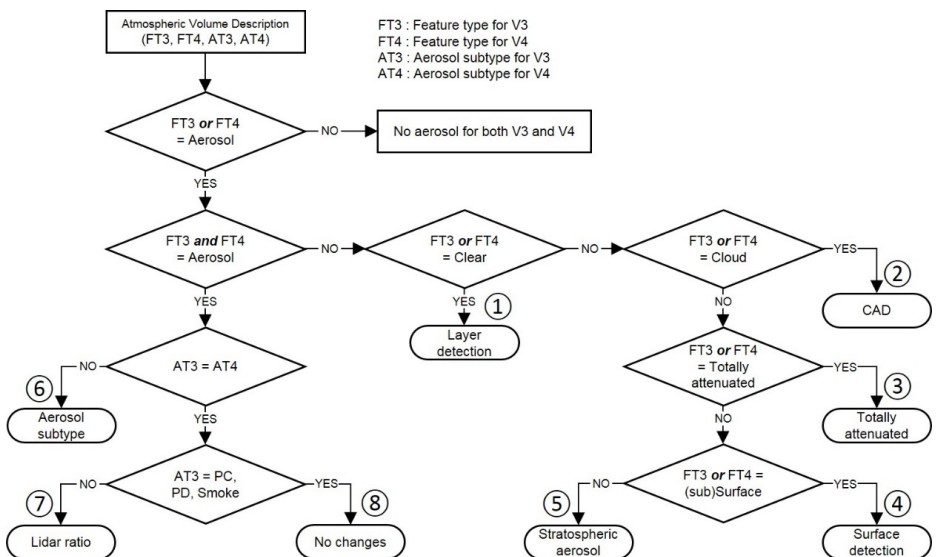

**Figure 14: Flowchart to categorize factors that impact on the AOD change between V3 and V4. Note that changes between polluted**
**continental and smoke are treated as no changes because the lidar ratios at 532 nm for those aerosols are same as 70 sr. FT3 and FT4 are feature types in V3 and V4. AT3 and AT4 are aerosol subtypes in V3 and V4.**

Table 6 quantifies the AOD changes from V3 to V4 for "all sky" conditions for the different factors categorized in Fig. 14; i.e., layer detection, CAD, surface detection, stratospheric aerosol, aerosol subtype, lidar ratio, and no change. Here, the all sky analysis

includes all profiles which contain identified aerosol regardless of the presence of clouds. All of the factors listed above contribute to the increase in AOD in V4 and the magnitudes of the AOD changes are strongly related to their occurrence frequencies (Table 6).



**Table 6: Mean column AOD changes (± standard deviation) from CALIOP V3 to V4 (defined as V4 – V3) and their bin frequencies for different reasons described in Fig. 5 for all-sky condition from 2007 to 2009.**

|  | Frequency (%) | | AOD change | |
|---|---|---|---|---|
|  | Night | Day | Night | Day |
| Layer detection | 20.9 | 17.5 | 0.007±0.050 | 0.005±0.061 |
| CAD | 11.7 | 12.2 | 0.007±0.159 | 0.014±0.121 |
| Total Attenuated | 1.7 | 1.3 | 0.008±0.075 | 0.003±0.047 |
| Surface detection | 1.5 | 1.5 | 0.001±0.025 | 0.002±0.015 |
| Stratospheric aerosol | 5.5 | 4.0 | 0.001±0.020 | 0.002±0.017 |
| Aerosol subtype | 16.5 | 18.7 | 0.003±0.081 | -0.004±0.068 |
| Lidar ratio | 31.5 | 36.8 | 0.013±0.067 | 0.012±0.060 |
| No change | 10.7 | 8.0 | 0.003±0.040 | 0.003±0.032 |
| Total (number of data) | 100 (868,893,575) | 100 (492,266,349) | 0.044±0.225 | 0.036±0.183 |

Global maps of AOD changes for each factor are shown in Fig. 15. CALIOP AOD has increased by 0.007 and 0.005 for nighttime

5    and daytime, respectively, because of changes in layer detection (Table 6 and Fig. 15(b)). This implies that the CALIOP V4 layer detection algorithm finds tenuous layers that were not found in V3. Note that no significant changes were made to the CALIOP layer detection algorithm in V4. The increased detection of faint layers is attributed primarily to changes in the 532 nm calibration coefficients that generally increase $\beta'$. The CAD algorithm classifies most of these new layers as aerosols.




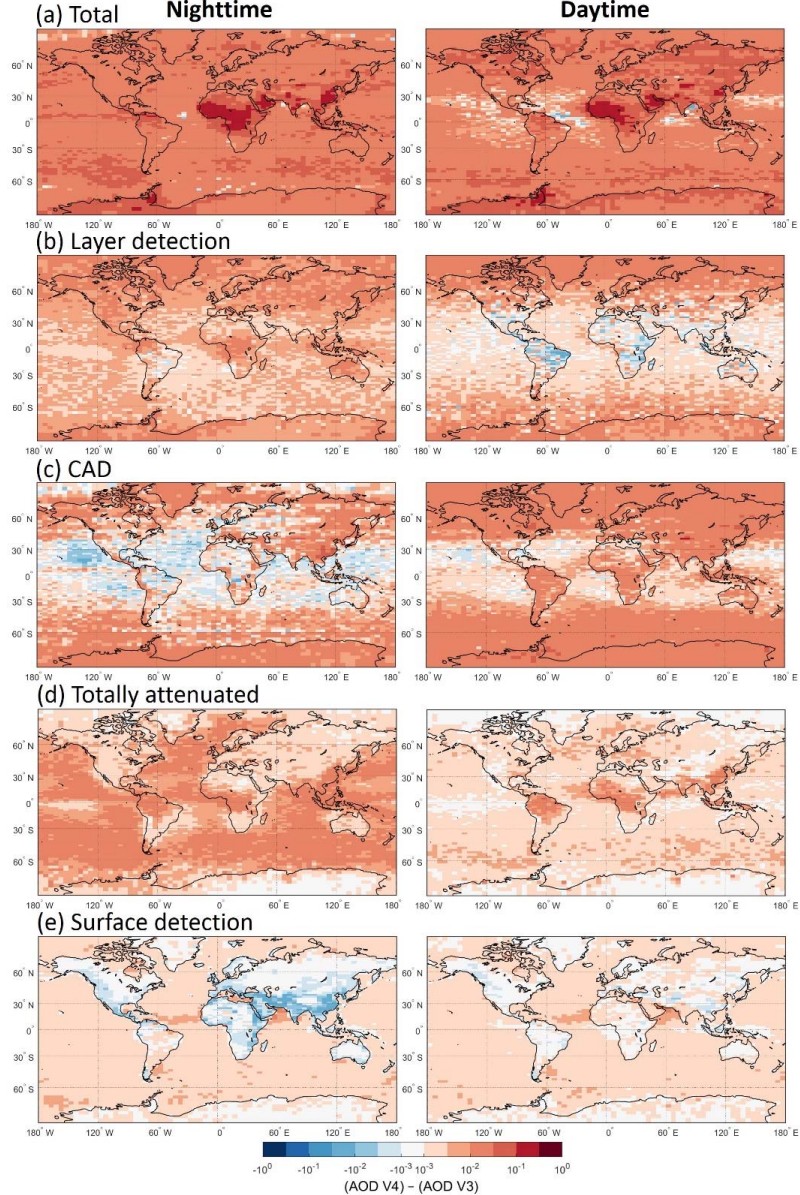

**Figure 15: Global maps of mean AOD differences between V3 and V4 for each factor categorized in Fig. 14 from 2007 to 2009; (a) total, (b) layer detection, (c) CAD, (d) totally attenuated, (e) surface detection, (f) stratospheric aerosol, (g) aerosol subtype, (h) lidar ratio, and (i) no changes. Left and right columns are for nighttime and daytime, respectively.**




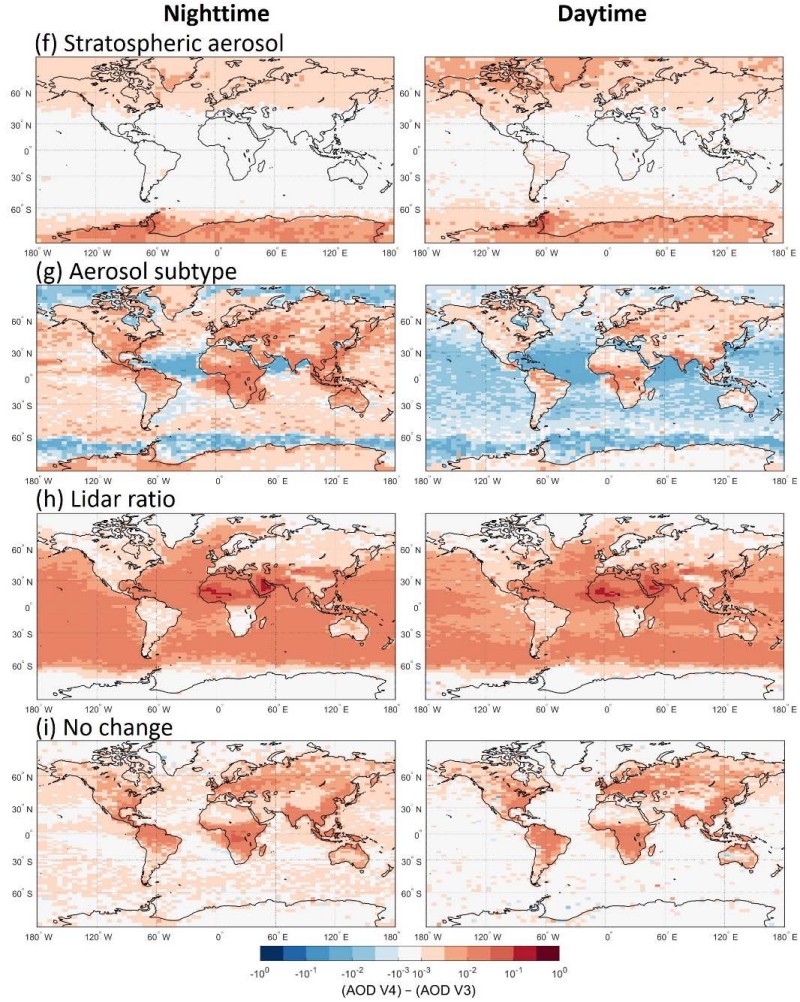

**Figure 15: (continued)**

5    Figure 15(c) shows that AOD changes due to CAD have a day and night difference. The mean daytime AOD increase is twice that for the nighttime (Table 6). The daytime AOD increase is due primarily to a net increase in the number of V4 aerosols. The number of new aerosols in V4 (i.e., layers that were classified as clouds in V3) is much larger (~3.4 times) than the converse (i.e., the aerosols in V3 that were classified as clouds in V4) at daytime. Among these new aerosols, the subtypes dust and polluted dust account for more than 50 %. The V4 CAD PDFs were deliberately tuned to be more sensitive to aerosol presence in the upper

10   troposphere and lower stratosphere, resulting in improved performance in distinguishing high altitude Asian dust plumes from cirrus (Liu et al., 2018). However, as a side effect, some fraction of the "cirrus fringes" detected along the edges and lower boundaries of cirrus clouds that were classified as clouds in V3 are now classified as aerosols in V4. Most of these fringes are subsequently identified as dust or polluted dust by the aerosol subtyping algorithm. These increases in misclassified dust and polluted dust at high altitudes contribute the most to the daytime AOD increase due to CAD. Although the misclassification of



cirrus fringes as aerosols also occurs in the V4 nighttime aerosol products, the gain and loss in the total aerosol number due to CAD are about the same and hence the change in nighttime aerosols cannot fully explain the increase in the nighttime mean AOD increase in Table 6. It appears that the change in V4 level 1 data calibration also plays an important role. As discussed below for Fig. 15(i), changes in the V4 level 1 data calibration alone can cause a nighttime AOD increase of 0.003, as shown in Table 6,

about half the nighttime AOD change of 0.007 due to CAD. However, the net change for each aerosol subtype varies largely and may play an important role in the regional AOD changes as seen in the left panel of Fig. 15(c).

If a bin is previously identified as totally attenuated in V3 and becomes aerosol in V4, the most likely cause is difference in surface detection. The V4 surface detection algorithm (Vaughan et al., 2018b) is considerably more effective than the V3 algorithm in

detecting the Earth's surface after penetrating atmospheric layers of substantial optical depth (e.g., cirrus clouds with optical depths of 2.5 or larger). As a result, regions where the signal was considered totally attenuated in V3 are now searched for the presence of features, and the aerosol layers detected in these regions contribute to an increase in AOD in V4 (Fig. 15(d)).

Changes in the surface detection and the newly introduced stratospheric aerosol types also contribute to AOD increases in V4, but

not significantly. Figure 15(e) shows that some surface signals misclassified as aerosols in V3 are correctly detected as surface in V4. This leads to a decrease in AOD especially in the southern Asian continent. Since the surface returns are much stronger than backscatter signals for aerosols, AOD changes appear relatively large. AOD increases due to newly introduced stratospheric aerosols in V4 are found mainly in the Arctic and Antarctic regions where the STS compositions of PSCs are ubiquitous in the polar winter and can be classified as stratospheric aerosol in V4 (Fig. 15(f)).

There is a decrease in mean of AOD due to aerosol subtype changes for daytime, but an increase for nighttime (Table 6). Figure 15(g) shows that AOD changes due to the aerosol type changes generally have opposite signs at day and night over oceans. The dominant aerosol type over oceans is clean marine, which has the smallest lidar ratio among the CALIOP aerosol models. Therefore, any changes from clean marine to other types of aerosol can lead to an AOD increase in V4. This is the dominant type

change over ocean for the nighttime. For the daytime, on the other hand, a type change from polluted dust to dusty marine occurs more frequently, as explained earlier (Sect. 3.2). The reduction in lidar ratio from polluted dust (55 sr) to dusty marine (37 sr) leads to a decrease in the mean daytime AOD. AODs decrease for both day and night over the mid-Atlantic and Indian Oceans as well as over the Arctic and Antarctic Oceans, as shown in Fig. 15(g). Dust can be frequently transported to the Atlantic and Indian Oceans from the Saharan and Arabian Deserts. Aerosol type changes from dust or polluted dust to dusty marine dominate in these

regions and lead to the AOD decreases in these regions. The AOD decreases over the Arctic and Antarctic Oceans are because the V4 aerosol subtyping algorithm now allows all aerosol types over these regions rather than solely the clean continental and polluted continental subtypes (Sect. 2.1.1).

Updates in lidar ratio led to an AOD increase of 0.013 and 0.012 for nighttime and daytime, respectively. The 532 nm lidar ratios

for three aerosol subtypes were changed in V4: clean marine, dust, and clean continental. Since the lidar ratios for these subtypes were all increased, AOD increased correspondingly. Figure 15(h) shows the AOD increases due to the lidar ratio updates. The AOD increase over oceans was due to the lidar ratio change for clean marine. The most significant AOD increase is seen over North Africa and the Arabian Peninsula where pure dust is a dominant aerosol type. The change was caused by the lidar ratio update for dust from 40 sr to 44 sr. Due to the nonlinear behavior of the AOD retrieval, this 10 % increase in lidar ratio yields an



increase of up to 20 % in AOD for dust layers as shown by AOD retrieval above opaque water clouds (Liu et al., 2015). Some of the AOD increases are even larger over North Africa and the Arabian Peninsula, where the dust is generally denser than that over the East Atlantic, thus further amplifying the nonlinear response of AOD to lidar ratio.

Figure 15(i) shows AOD changes even if there are no changes in aerosol subtype and lidar ratio. AOD is slightly increased especially over land. These changes are due to differences in calibration and two-way transmittance for upper layers between V3 and V4.

AOD changes between V3 and V4 for "cloud-free" conditions are typically smaller than in the all-sky cases discussed above.
Cloud-free cases are restricted to those profiles where no clouds were detected for both V3 and V4 from the level 2 5 km profile product. Table 7 shows occurrence frequencies and AOD changes for cloud-free conditions. The frequencies for layer detection are substantially reduced compared to all-sky. This implies that, when compared to V3, the V4 aerosol detection frequency is higher in cloudy-skies but relatively unchanged for clear-skies. This behavior is not unexpected, and can be at least partially explained by the improved surface detection in V4 that identifies more transparent air columns compared with V3. This behavior
can also be attributed partially that the presence of more misclassified cirrus fringes in V4 (Liu et al., 2018). Comparing Tables 6 and 7, a larger fraction of aerosols maintain their subtype in the cloud-free conditions than in the all-sky conditions (i.e., the lidar ratio category in Table 7 is 45.9 % night and 50.7 % day vs. 30.2 % night and 35.9 % day in Table 6). Due to the largely increased frequencies, AOD increases by the lidar ratio updates in V4 overwhelm all the other factors for cloud-free skies compared to cloudy skies.

**Table 7: Same as Table 6 but for cloud-free sky.**

| | Frequency (%) | | AOD change | |
| --- | --- | --- | --- | --- |
| | Night | Day | Night | Day |
| Layer detection | 17.3 | 13.2 | 0.006±0.030 | 0.003±0.042 |
| CAD | - | - | - | - |
| Total Attenuated | 0.1 | 0.2 | 0.000±0.014 | 0.001±0.022 |
| Surface detection | 2.2 | 2.1 | 0.001±0.029 | 0.003±0.014 |
| Stratospheric aerosol | 2.9 | 1.6 | 0.001±0.009 | 0.001±0.008 |
| Aerosol subtype | 18.2 | 22.8 | 0.000±0.076 | -0.008±0.071 |
| Lidar ratio | 46.1 | 50.6 | 0.016±0.056 | 0.013±0.057 |
| No change | 13.2 | 9.7 | 0.003±0.030 | 0.002±0.028 |
| Total (number of data) | 100 (362,664,890) | 100 (249,431,801) | 0.027±0.113 | 0.015±0.115 |

**4 Preliminary validation with AERONET and MODIS**

A low bias of the V3 CALIOP AOD estimates relative to both MODIS and AERONET has been established in a number of previous publications (Kacenelenbogen et al., 2011; Oo and Holz, 2011; Redemann et al., 2012; Schuster et al., 2012; Kim et al.,
2013; Omar et al., 2013). In this study, CALIOP AODs for both V3 and V4 are compared with AERONET and MODIS using collocation methods similar to Omar et al. (2013) and Kim et al. (2013). The collocation criteria adopted for CALIOP and AERONET (level 2) require data acquisition times within ±30 min and spatial matching of the CALIOP footprint to within a 40 km radius of the AERONET site. Enforcing these criteria yields 736 data pairs from 176 sites over the globe from 2007 to 2009.



MODIS level 2 AOD and CALIOP level 2 column integrated AODs whose distance from the center of MODIS grid is less than 10 km are selected as collocated data pairs for the same period. CALIOP level 2 data with extinction QC flags of 0, 1, and 16 are used for both V3 and V4. MODIS collection 6 Dark-Target AODs (Levy et al., 2013), "Effective_Optical_Depth_Average_Ocean" over ocean and "Corrected_Optical_Depth_Land" over land at 550 nm, with "Quality_Assurance_Ocean" of 1 (marginal) or higher

and "Quality_Assurance_Land" of 3 (very good) are used for comparison. To remove cloud contamination, data pairs with CALIOP cloud column optical depths greater than 0 or MODIS cloud fractions greater than 0 % are rejected.

Global maps of AOD differences between CALIOP and AERONET/MODIS (CALIOP – AERONET/MODIS) are shown in Fig. 16. The color-coded maps show AOD differences relative to MODIS, while differences relative to AERONET are shown as

individual filled circles on the map. The AOD differences between CALIOP and AERONET are generally similar between the two versions. Likewise, regional CALIOP – MODIS AOD differences over oceans are generally similar for V3 and V4, except for the southern oceans (< 30° S), where the V4 AOD differences are slightly larger. Another noticeable difference between the two versions is that the AOD difference is reduced in V4 off the southwest African coast. This is mostly related to aerosol type changes from correcting the classifications of elevated smoke plumes previously misclassified as clean marine over this region in V3 (Sect.

2.3). Over land, AOD differences relative to MODIS typically increase in V4 compared to V3. The increases over tropical and southern Africa are, in part, due to corrections in $\delta_p^{est}$ which were overestimated in V3 (Sect. 2.1.2). These corrections tend to change aerosol subtype classifications to aerosol subtypes with higher lidar ratios (e.g., dust to polluted dust). Differences between CALIOP and MODIS are most noticeable in Africa and South Asia, whereas agreement with AERONET in these regions tends to be much better.

Resolution of the inconsistency between the comparisons with AERONET and MODIS points to the need for further validation studies, especially over land. Since the MODIS over-ocean algorithm is generally more accurate than over-land (Levy et al., 2013), AOD differences between CALIOP and MODIS over land are excluded from further consideration in our analyses. Global mean and median of AOD differences between CALIOP and AERONET/MODIS for V3 and V4 are shown in Table 8. Both V3 and V4

show that the mean CALIOP AODs are smaller than AERONET and MODIS (ocean), but the mean (median) discrepancies are reduced from -0.064 (-0.052) to -0.051 (-0.045) for AERONET and from -0.010 (-0.012) to -0.006 (-0.008) for MODIS over ocean. The CALIOP V3 bias over global oceans relative to MODIS collection 6 is smaller compared to previous studies (e.g., Oo and Holz, 2011; Redemann et al., 2012; Kim et al., 2013). These earlier studies used MODIS collection 5 and global AOD for MODIS collection 6 has decreased by 0.02 over ocean compared to collection 5 (Levy et al., 2013).

**Table 8: Mean (± standard deviation) and median (± median absolute deviation) of AOD difference between CALIOP and AERONET/MODIS (defined as CALIOP – AERONET/MODIS) for V3 and V4 from 2007 to 2009.**

|  | CALIOP V3 | | CALIOP V4 | | Number of data pairs |
| --- | --- | --- | --- | --- | --- |
|  | mean | median | mean | median | |
| AERONET | -0.064±0.087 | -0.052±0.028 | -0.051±0.085 | -0.045±0.025 | 736 |
| MODIS (Ocean) | -0.010±0.070 | -0.012±0.025 | -0.006±0.068 | -0.008±0.025 | 911,376 |
| MODIS (Land) | 0.069±0.195 | 0.062±0.091 | 0.121±0.225 | 0.090±0.098 | 38,142 |



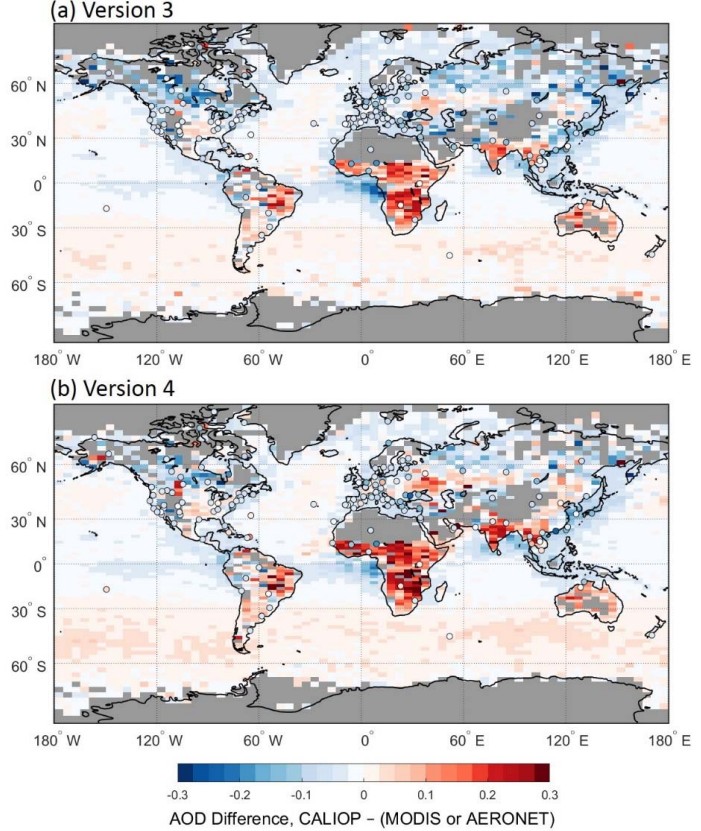

**Figure 16: Global maps of mean AOD difference between CALIOP and MODIS. CALIOP data for (a) V3 and (b) V4 and MODIS collection 6 from 2007 to 2009. Mean AOD difference between CALIOP and AERONET are shown in circles.**

5    Figure 17 compares the V3 and V4 CALIOP AOD differences with respect to MODIS over ocean from 2007 to 2009. Points on the one to one line (black dotted) correspond to no AOD change between V3 and V4. Points closer to the x-axis (y=0) represent AOD biases that are smaller in V4 compared to V3, while points closer to the y-axis (x=0) represent AOD biases that are larger in V4. The linear trend line has a slope less than 1 which means that the overall AOD bias is reduced in V4 (perfect agreement between V4 and MODIS would yield a slope of zero.). AOD biases with respect to MODIS for V3 and V4 have a distribution

10    close to the one to one line. The slope of the linear trend is 0.82, indicating a small reduction in the AOD bias from V3 to V4.





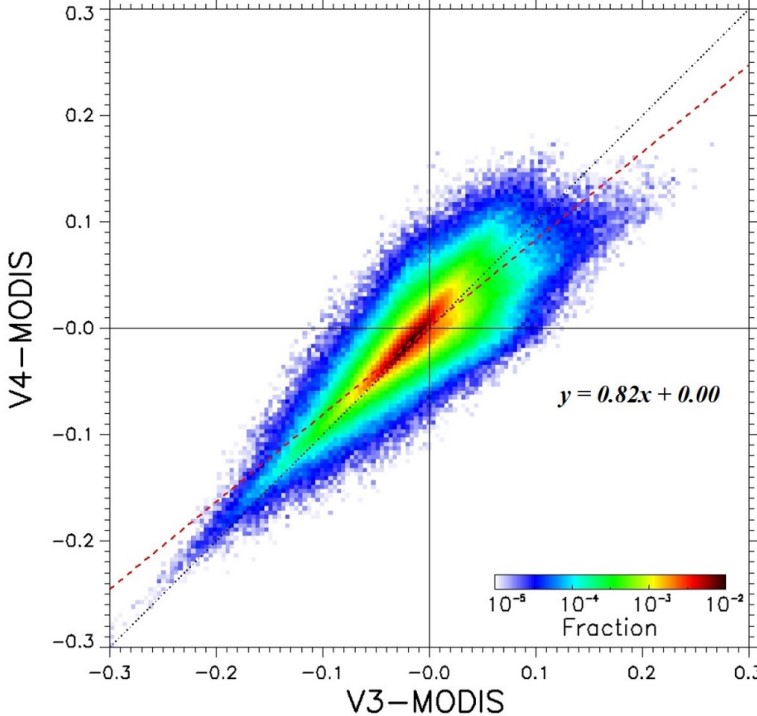

**Figure 17: Distribution of V3 CALIOP AOD difference from MODIS versus V4 CALIOP AOD difference over ocean from 2007 to 2009. A linear regression line is shown in red dashed line with equation.**

## 5 Summary

The CALIPSO version 4.10 (V4) lidar level 2 data products were released in November 2016. V4 is the first wholly new set of data products since the initial release of the Version 3 (V3) series of products in May 2010. Algorithm updates and data product changes for V4 aerosol subtyping algorithms are discussed in this study. The most significant algorithm updates in V4 are as follows.

• All aerosol subtypes are now allowed over snow, ice, and tundra surfaces, whereas only clean continental and polluted continental aerosols were allowed in previous versions.

• A new aerosol subtype, dusty marine, has been introduced. The wide-spread occurrence of layers misclassified as polluted dust over the ocean in previous versions has been rectified, and these layers are now correctly classified as dusty marine.

• The polluted continental and smoke aerosol types in previous versions have been renamed in V4 to "polluted
continental/smoke" and "elevated smoke", respectively.

• A new scheme, the Subtype Coalescence Algorithm for AeRosol Fringes (SCAARF), is applied to re-evaluate the aerosol subtype of aerosol layers detected at coarse resolutions below overlying horizontally adjacent layers.

• Stratospheric aerosol subtypes have been introduced for ash, sulfate/other, smoke and polar stratospheric aerosol.

• Aerosol lidar ratios have updated for clean marine, dust, clean continental, and elevated smoke to represent the current state of
knowledge for these types.



Feature type and aerosol subtype changes between V3 and V4 are investigated. Tropospheric aerosol occurrence frequency has increased by 18 % in V4 compared to V3, which implies that the V4 algorithm detects more weakly scattering layers that are subsequently classified as aerosols. Moreover, including stratospheric aerosols in V4, aerosol occurrence frequency increases by

31 % relative to V3. As consequence, the occurrence frequency of most aerosol subtypes also increases. The sole exceptions are polluted dust and clean continental. Layers previously classified as polluted dust that have base altitudes less than 2.5 km are now classified as dusty marine in V4. Similarly, the clean continental aerosols that were ubiquitous in the polar regions in the V3 and earlier data sets are now classified as other subtypes in V4.

The CALIOP level 2 mean column integrated AOD at 532 nm has increased by 0.044 for nighttime and 0.036 for daytime for all sky in V4 from 2007 to 2009. The most significant reasons for the AOD increase in V4 are changes in lidar ratio, CAD and layer detection. For cloud-free skies, however, the contribution of CAD and aerosol layer detection is not as pronounced. Initial comparison of AERONET and MODIS with both versions of CALIOP shows that mean AOD differences with AERONET and MODIS (ocean) are reduced in V4 compared to V3. However, the CALIOP AOD estimates remain low relative to MODIS, and

this disparity will not be reconciled based solely on future modifications to the CALIOP aerosol typing and lidar ratio selection algorithms. MODIS makes multi-spectral total column measurements from which AOD estimates can be derived, but cannot provide height-resolved estimates of extinction. In principal, CALIOP has the capacity to deliver these height-resolved estimates of aerosol extinction coefficients on a global scale. But, to date, CALIOP has limited the retrieval of aerosol optical properties to those regions where the layer detection algorithm and cloud-aerosol discrimination algorithm have positively identified the

presence of aerosol in the atmosphere. No attempt is currently being made to retrieve aerosol optical properties in those regions where the aerosol loading lies below the CALIOP detection limits, and hence many of the differences seen between the CALIOP and MODIS estimates of AOD should be expected. Should the CALIOP retrieval strategy change in future data releases, comparisons with MODIS will need to be thoroughly and rigorously re-examined.

### *Acknowledgements*

The CALIPSO data used in this study were obtained from the NASA Langley Atmospheric Science Data Center (https://eosweb.larc.nasa.gov/project/calipso/calipso_table). The MODIS data were obtained through the NASA Goddard Space and Flight Center Data Center (GSFC) Atmosphere Archive & Distribution System (https://ladsweb.nascom.nasa.gov/). We thank the AERONET Principal Investigators and their staff for establishing and maintaining the AERONET at the 176 sites used in this investigation. Man-Hae Kim was supported by a NASA Postdoctoral Program Fellowship.

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
