# Peer review of "The CALIPSO Version 4 Automated Aerosol Classification and Lidar Ratio Selection Algorithm"

_Atmospheric Measurement Techniques, 2018_

## Referee Comment (RC1) · Anonymous Referee #1 · 25 Jul 2018

Review: The CALIPSO Version 4 Automated Aerosol Classification and Lidar Ratio Selection Algorithm

Authors: Man-Hae Kim, Ali H. Omar, Jason L. Tackett, Mark A. Vaughan, David M. Winker, Charles R. Trepte, Yongxiang Hu, Zhaoyan Liu, Lamont R. Poole, Michael C. Pitts, Jayanta Kar, and Brian E. Magill

General comments:

The document clearly describes the steps taken in the algorithm and the changes made between the different versions, in that sense it is a very readable version of the ATBD and will be important for the users of the data to read. The paper is very lengthy but that is by no means a bad thing for this type of article which has to describe all the

changes and the impacts of each.

I was very lucky to have a pro-active editor who made a large number of substantial comments resulting in a revised version before I had to go through the paper, making the task a lot easier. I will not add comments from my side which touches those discussions, knowing that some of the specific details have been looked at in detail. For the remainder the authors only need to look at some minor comments from my side.

I personally very much like the change in the concept in V4 of retrieving the subclasses for stratospheric aerosols and allowing the more general aerosols over polar regions. The main part which I am still wondering about is why the 1064nm channel provides you with no additional information in the troposphere. It is mentioned that within the PBL the color ratio does not help with the classification, however I did not notice a similar remark for the remainder of the troposphere. Please include a small discussion on why the color ratio has no impact on the classification in the troposphere since now only 2 of the three signals are used in this regime.

In a lot of places within the document the effects of the new Earth's Surface detection scheme (Vaughan 2018b) is discussed. Sadly enough I do not have a draft version available of this paper, making it impossible to understand the why's of any changes. There is enough information provided to 'trust' the results, so I will do without for this review.

Minor comments:

Page 2, Line 25: compare –> compared Line 26: Even though it is obvious please add V3 to CALIOP AOD

Page 6, Line 12: AOD differences

Page 10, Figure 5: would it be possible to overplot the Arctic PSC season (winter that year) to show that the -70 is consistent for both absolute high latitude regions.

Page 14, Line 7: two releases. Also add a bit more information/origin on the 20 and

80km resolution

Page 20, Line 4: demonstrates an

Page 21, Line 20: Can you explain the 31% quoted there. My guess is that it is the normalized values of 1.18 and 0.13 combined, but a bit more explanation here would make it a lot easier to comprehend

Page 22, Line 7: Add what the averaging boxes are 1 deg x 1deg ?

Page 24, Line 2 : anan Line 13: give full name of AVD (maybe I missed it earlier) Line 18: WhileWhile

Page 24 Line19 & Page 30 line 15 : Remove correctly from 'correctly classified & correctly detected'. The word correct should not appear in a this text as the absolute truth is not available. You can maybe use 'more realistically' or something like that if you would like to add an adverb. I agree that for surface detection one could consider it, however in the previous version the surface was also thought to be correct.

Page 24 Line24: Rewrite sentence 'Similarly . . ..in V3'.

Page 25,Table 5. Most of the work is a comparison between the two versions where we now have absolute values in this table. What is clearly noticeable is not only the change in mean value but also the larger standard deviations for the V4. Since we are looking at the 0, 1, 16 and 18 QC flags only, I would like to see the subdivision of the mean and std deviations for these four individual flagged pixels and see where the std deviation has increased most and a small discussion why. Please add the values for the four QC flags individually to the table.

Page 30 Line 21: a decrease in the mean AOD

Page 32 Lines 27-29: The discussion is hard to follow, mostly due to the word smaller (I think). I guess you mean smaller in absolute value here or not. In any case please rephrase this sentence to make it crystal clear what you mean

Page 34, Line 13: are now correctly classified as → are more realistically classified as Line 19 : have been updated Figure 17: By eye it looks like there are two distributions combined (heart shaped distribution) with one above and the other below the 1-1 line. Is there a reason for this, one day and the other night?

Figures: The axes font-sizes of Figures 2, 3, 7, 8, 9, 11, 15 are extremely small, I think it would improve reading/glancing of the figures enormously if these would be increased.

---

## Referee Comment (RC2) · Anonymous Referee #2 · 2 Aug 2018

The authors have put together a very well-written and easy-to-follow description of the CALIPSO Version 4 automatic aerosol classification and lidar ratio selection algorithm. There have been some significant changes from the previous version including: updates to the lidar ratios for several aerosol types, the inclusion of a new dusty marine subtype, full use of the aerosol classification algorithm over all surfaces, aerosol classification in the stratosphere, and the addition of a new algorithm, SCAARF, to homogenize aerosol types when "fringes" occur. Additionally, the manuscript validates the updated algorithm by showing how the resulting AOD is in closer agreement with both MODIS and AERONET.

The authors must be applauded for presenting such a strong manuscript, that there is not much to be asked in way of revisions. If I were to make one complaint (and I

won't... at least not fully) it is the use of 2.5 km as a standard approximation for the planetary boundary layer without regard to latitude, surface, time of day, or season. Understandably, there are some practical reasons for doing so, and the decision is based on results from published literature and largely consistent with Version 3. Each change in the aerosol classification and lidar ratio selection algorithm from V3 to V4 is justified and appropriately referenced based on the most current understanding within the aerosol community. The manuscript should be published after some very minor technical corrections.

Page 6, Line 12: "AODdifferences" should be "AOD differences"

Figure 3: Upon my first inspection of this figure, I was drawn to the differences in the number of aerosol retrievals both within and without the white dashed ellipse. Fortunately, the causes of these differences are thoroughly discussed later in section 3.3. However, as a courtesy for the inquisitive reader, it would be nice to add a sentence to the text informing that this topic will be addressed later in the manuscript.

Page 11, Line 20: "aa" should be "a"

Page 12, Line4: .. should be .

Page 14, Line7: "releasesreleases" should be "releases"

Page 15, Line 22: ;; should be ;

Page 15, Line 22: The overlines should be removed for the fringe variables delta and chi to match the format in equation 3

Page 20, Line 4: "demonstratesan" should be "demonstrates an"

Page 24, Line 18: "WhileWhile" should be "While"

Page 25, Line 21: "subtypes in reported in" should be "subtypes reported in"

---

## Referee Comment (RC3) · Anonymous Referee #3 · 15 Aug 2018

This paper introduces CALIPSO V4 updates in the L2 aerosol subtyping algorithms, investigates the resulting AOD differences between V3 and V4, and compares CALIOP AOD with AERONET and MODIS. This paper adequately summarizes information on the updates of the algorithms and characteristics of the products, includes useful information on them, and is well written. Thus this paper will be published after some minor revisions.

(1) Title of this paper seems to be inadequate. The key topic of this paper is to evaluate the algorithm to create the Version 4 L2 subtyping products by comparing with Version 3 products. The current title rather gives us impression that this study developed the algorithm .

(2) "Section 2" Section 2 seems to be too long and may mislead. The key part of this

paper is thought to be section 3 (and section 4), however, section 2, which explains the updates of the algorithms and products and is not essential to this paper, looks the key of this paper.

(3) minor comments and typos P2 L6: "(AOD" => (AOD)

P11 L20: "aa" => "a"

P14 Fig 9 (b): X-axis of Fig. (b) seems different from the other figures (a and b).

P14 L7 "releasereleases" => release

P18 L13: "Based on an assumed external mixture of ∼∼∼(65:35 by surface area)" What is 65:35? You should explain more on this point.

P21 Table 4: This table is very complicated and it is difficult to understand it. You need to explain more to let readers understand this table.

P24 L15: "WhileWhile"

P32 Table 8 :Are the values of the MODIS (Land) negative? (e.g., 0.069 => -0.069)
* * *

---

## Author Comment (AC1) · 15 Sep 2018

We thank the referee for their careful reading of our manuscript and their thoughtful comments. We have reproduced the referee's comments below (in black) and included our responses in-line (in blue).

Review: The CALIPSO Version 4 Automated Aerosol Classification and Lidar Ratio Selection Algorithm

Authors: Man-Hae Kim, Ali H. Omar, Jason L. Tackett, Mark A. Vaughan, David M. Winker, Charles R. Trepte, Yongxiang Hu, Zhaoyan Liu, Lamont R. Poole, Michael C. Pitts, Jayanta Kar, and Brian E. Magill

**General comments:**
The document clearly describes the steps taken in the algorithm and the changes made between the different versions, in that sense it is a very readable version of the ATBD and will be important for the users of the data to read. The paper is very lengthy but that is by no means a bad thing for this type of article which has to describe all the changes and the impacts of each.

I was very lucky to have a pro-active editor who made a large number of substantial comments resulting in a revised version before I had to go through the paper, making the task a lot easier. I will not add comments from my side which touches those discussions, knowing that some of the specific details have been looked at in detail. For the remainder the authors only need to look at some minor comments from my side.

I personally very much like the change in the concept in V4 of retrieving the subclasses for stratospheric aerosols and allowing the more general aerosols over polar regions. The main part which I am still wondering about is why the 1064nm channel provides you with no additional information in the troposphere. It is mentioned that within the PBL the color ratio does not help with the classification, however I did not notice a similar remark for the remainder of the troposphere. Please include a small discussion on why the color ratio has no impact on the classification in the troposphere since now only 2 of the three signals are used in this regime.
> The CALIPSO scene classification algorithm uses 1064 nm channel for cloud and stratospheric aerosols but not for tropospheric aerosol. It is mainly because low signal to noise ratio (SNR) for 1064 nm especially for aerosol layers during daytime. It might be available to use 1064 nm for some dense aerosol layer. But, the returned lidar signal is relatively weak for (faint) aerosols compared to cloud. In order to apply uniform criteria for all detected aerosol layers, the algorithm does not use 1064 nm for aerosol subtyping. We added some comments on this in the text (Sec. 2.1).

In a lot of places within the document the effects of the new Earth's Surface detection scheme (Vaughan 2018b) is discussed. Sadly enough I do not have a draft version available of this paper, making it impossible to understand the why's of any changes. There is enough information provided to 'trust' the results, so I will do without for this review.

**Minor comments:**
Page 2, Line 25: compare –> compared Line 26: Even though it is obvious please add V3 to CALIOP AOD
> It is corrected.

Page 6, Line 12: AOD differences
> It is corrected.

Page 10, Figure 5: would it be possible to overplot the Arctic PSC season (winter that year) to show that the -70 is consistent for both absolute high latitude regions.
> We updated figure 5 for the Arctic and Antarctic PSC seasons.

Page 14, Line 7: two releases. Also add a bit more information/origin on the 20 and 80km resolution
> It is corrected and simple explanation and reference are added.

Page 20, Line 4: demonstrates an
> It is corrected.

Page 21, Line 20: Can you explain the 31% quoted there. My guess is that it is the normalized values of 1.18 and 0.13 combined, but a bit more explanation here would make it a lot easier to comprehend
> We added 13% of newly introduced stratospheric aerosol in the text.

Page 22, Line 7: Add what the averaging boxes are 1 deg x 1deg ?
> We compared profile to profile for every Level 2 aerosol profile products which have 5 km x 60 m (5 km x 180 m for above 20.2 km) range bin. It is described in the second paragraph of Sec. 3.

Page 24, Line 2 : anan Line 13: give full name of AVD (maybe I missed it earlier) Line 18: WhileWhile
> They are corrected. The full name of AVD is introduced earlier in Sec. 3.

Page 24 Line19 & Page 30 line 15 : Remove correctly from 'correctly classified & correctly detected'. The word correct should not appear in this text as the absolute truth is not available. You can maybe use 'more realistically' or something like that if you would like to add an adverb. I agree that for surface detection one could consider it, however in the previous version the surface was also thought to be correct.
> They are changed from 'correctly' to 'realistically'.

Page 24 Line24: Rewrite sentence 'Similarly . . ..in V3'.
> The sentence is rewritten to make it clear.

Page 25,Table 5. Most of the work is a comparison between the two versions where we now have absolute values in this table. What is clearly noticeable is not only the change in mean value but also the larger standard deviations for the V4. Since we are looking at the 0, 1, 16 and 18 QC flags only, I would like to see the subdivision of the mean and std deviations for these four individual flagged pixels and see where the std deviation has increased most and a small discussion why. Please add the values for the four QC flags individually to the table.
> We note that the standard deviations (STDs) in table 5 are not related with their uncertainties. They show broadness (shape) of AOD distribution. Larger standard deviations (STDs) in V4 are strongly related to the AOD increase. If AODs increase with same amount for all data points, STD remains unchanged in V4 (only the mean will increase). However, if all AODs increase by 10%, for example, STD should also increase by 10%. (If $\sigma$ is STD for $x_i$ (i=0,1,2…n), STD for a $x_i$ + b is a$\sigma$.) Below table shows % difference from V3 to V4 for both mean and STD. As shown in the table, the % differences for mean and STD are comparable. We thought it is expectable and there is no specific relation with QC flags for STD increase in V4. Thus, we didn't modify the text.

| | Mean | | Standard Deviation | |
|---|---|---|---|---|
| | Night | Day | Night | Day |
| V3 | 0.084 | 0.090 | 0.162 | 0.150 |
| V4 | 0.128 | 0.126 | 0.242 | 0.202 |
| Increased in % | 52.4 | 40.0 | 49.4 | 34.7 |

Also, more detailed description about aerosol QC flags can be found in Young et al. (2018), another paper of our AMT special issue about the CALIPSO Version 4 updates. In Table 3(b) from Young et al. (2018), the majority of QC flags falls into QC=0 (95.32% for V3 and 92.97% for V4).
*Young, S. A., Vaughan, M. A., Garnier, A., Tackett, J. L., Lambeth, J. B., and Powell, K. A.: Extinction and Optical Depth Retrievals for CALIPSO's Version 4 Data Release, Atmos. Meas. Tech. Discuss., https://doi.org/10.5194/amt-2018-182, in review, 2018.*

Page 30 Line 21: a decrease in the mean AOD
> It is corrected.

Page 32 Lines 27-29: The discussion is hard to follow, mostly due to the word smaller (I think). I guess you mean smaller in absolute value here or not. In any case please rephrase this sentence to make it crystal clear what you mean
> The sentence is modified to make it clear.

Page 34, Line 13: are now correctly classified as ! are more realistically classified as Line 19 : have been updated
> It is corrected.

Figure 17: By eye it looks like there are two distributions combined (heart shaped distribution) with one above and the other below the 1-1 line. Is there a reason for this, one day and the other night?

> The reviewer's comment is correct. The core (heart shaped distribution) is nearly parallel to 1:1 line, just above it. And minor branch is below (and right) the 1:1 line with the slope of <1. The former is mainly due to increase of the lidar ratio for clean marine and dust. As shown in Figure 13, (clean marine + dust) counts for ~60% of whole aerosol layers. Due to the lidar ratio increase for these two aerosols, CALIOP V4 AOD increases and the heart shaped distribution in Figure 17 located just above the 1:1 line. The latter corresponds mainly to dusty marine aerosol in V4. The most of dusty marine aerosols come from polluted dust in V3. Because the lidar ratios for polluted dust and dusty marine are 55 sr and 37 sr, respectively, V4 AOD decreases, shown as the minor branch in Figure 17.

This is a simple description, but in fact, it is much more complicated. We compared every single MODIS pixel with collocated CALIPSO data points, which means each collocated data pair is consist with one MODIS data pixel and 3~5 CALIPSO data points. Therefore, multiple aerosol layers with different types may coexist. Aerosol types may change from one to another or a new aerosol layers can be detected in V4. Because we averaged column AOD for collocated CALOPSO data points to compare with MODIS, it is difficult to separate those different contributions in Figure 17. Many figures and tables with additional analysis are needed for this discussion, and it is beyond the main topic of this paper.

Figures: The axes font-sizes of Figures 2, 3, 7, 8, 9, 11, 15 are extremely small, I think it would improve reading/glancing of the figures enormously if these would be increased.

> We adjusted figure sizes and font sizes.

---

## Author Comment (AC2) · 15 Sep 2018

We thank the referee for their careful reading of our manuscript and their thoughtful comments. We have reproduced the referee's comments below (in black) and included our responses in-line (in blue).

The authors have put together a very well-written and easy-to-follow description of the CALIPSO Version 4 automatic aerosol classification and lidar ratio selection algorithm. There have been some significant changes from the previous version including: updates to the lidar ratios for several aerosol types, the inclusion of a new dusty marine subtype, full use of the aerosol classification algorithm over all surfaces, aerosol classification in the stratosphere, and the addition of a new algorithm, SCAARF, to homogenize aerosol types when "fringes" occur. Additionally, the manuscript validates the updated algorithm by showing how the resulting AOD is in closer agreement with both MODIS and AERONET.

The authors must be applauded for presenting such a strong manuscript, that there is not much to be asked in way of revisions. If I were to make one complaint (and I won't... at least not fully) it is the use of 2.5 km as a standard approximation for the planetary boundary layer without regard to latitude, surface, time of day, or season. Understandably, there are some practical reasons for doing so, and the decision is based on results from published literature and largely consistent with Version 3. Each change in the aerosol classification and lidar ratio selection algorithm from V3 to V4 is justified and appropriately referenced based on the most current understanding within the aerosol community. The manuscript should be published after some very minor technical corrections.

Page 6, Line 12: "AODdifferences" should be "AOD differences"
> It is corrected.

Figure 3: Upon my first inspection of this figure, I was drawn to the differences in the number of aerosol retrievals both within and without the white dashed ellipse. Fortunately, the causes of these differences are thoroughly discussed later in section 3.3. However, as a courtesy for the inquisitive reader, it would be nice to add a sentence to the text informing that this topic will be addressed later in the manuscript.
> We added a sentence.

Page 11, Line 20: "aa" should be "a"
Page 12, Line4: .. should be .
Page 14, Line7: "releasesreleases" should be "releases"
Page 15, Line 22: ;; should be ;
Page 15, Line 22: The overlines should be removed for the fringe variables delta and chi to match the format in equation 3
Page 20, Line 4: "demonstratesan" should be "demonstrates an"
Page 24, Line 18: "WhileWhile" should be "While"
Page 25, Line 21: "subtypes in reported in" should be "subtypes reported in"

> They are all corrected.

---

## Author Comment (AC3) · 15 Sep 2018

We thank the referee for their careful reading of our manuscript and their thoughtful comments. We have reproduced the referee's comments below (in black) and included our responses in-line (in blue).

This paper introduces CALIPSO V4 updates in the L2 aerosol subtyping algorithms, investigates the resulting AOD differences between V3 and V4, and compares CALIOP AOD with AERONET and MODIS. This paper adequately summarizes information on the updates of the algorithms and characteristics of the products, includes useful information on them, and is well written. Thus this paper will be published after some minor revisions.

(1)Title of this paper seems to be inadequate. The key topic of this paper is to evaluate the algorithm to create the Version 4 L2 subtyping products by comparing with Version 3 products. The current title rather gives us impression that this study developed the algorithm.

> This paper is submitted as a part of the AMT special issue, "CALIPSO version 4 algorithms and data products" which introduces new algorithm updates for the CALIPSO version 4 products. The primary purpose is to introduce the new elements and refinements to the V4 algorithm that were developed by this study. The current tile is also consistent with the title of Omar et al. (2009) which introduce the aerosol subtyping algorithm for the previous version (V3). For these reasons, we did not change the title.

(*Omar, A., D. Winker, C. Kittaka, M. Vaughan, Z. Liu, Y. Hu, C. Trepte, R. Rogers, R. Ferrare, R. Kuehn, C. Hostetler, 2009: "The CALIPSO Automated Aerosol Classification and Lidar Ratio Selection Algorithm", J. Atmos. Oceanic Technol., 26, 1994-2014, doi:10.1175/2009JTECHA1231.1.*)

(2) "Section 2" Section 2 seems to be too long and may mislead. The key part of this paper is thought to be section 3 (and section 4), however, section 2, which explains the updates of the algorithms and products and is not essential to this paper, looks the key of this paper.

> Section 2 is key topic of this study rather than Sections 3 and 4 as mentioned above. Sections 3 and 4 evaluate the consequences of changes described in Section 2 and seeks to demonstrate that improvements were made. We think that the order and importance of section 2 is fine the way it is. We slightly modified the last paragraph in Section 1 to make the purpose of the paper clear.

(3) minor comments and typos

P2 L6: "(AOD" => (AOD)

> It is corrected.

P11 L20: "aa" => "a"

> It is corrected.

P14 Fig 9 (b): X-axis of Fig. (b) seems different from the other figures (a and b).

> It is corrected.

P14 L7 "releasereleases" => release

> It is corrected.

P18 L13: "Based on an assumed external mixture of ~~~(65:35 by surface area)" What is 65:35? You should explain more on this point.

> We added "mixing ratio of 65:35" instead of "65:35".

P21 Table 4: This table is very complicated and it is difficult to understand it. You need to explain more to let readers understand this table.

> We think the title of the table explains fully about the table.

P24 L15: "WhileWhile"

> It is corrected.

P32 Table 8 :Are the values of the MODIS (Land) negative? (e.g., 0.069 => -0.069)
> Positive values are correct. The mean CALIOP AOD is larger than the mean MODIS AOD over land for both V3 and V4 in this study. Large differences over the mid and south Africa (CALIOP is much larger as shown in Figure 16) are responsible for these results. But as mentioned in the text, the values in Table 8 are averaged values for all collocated data points between CALIOP and MODIS. They are not the mean values for gridded-averaged AOD differences. Further studies for the validation should be followed as mentioned in the last paragraph of Conclusion.

---

## Referee Comment (RC4) · Anonymous Referee #4 · 19 Sep 2018

Review of "The CALIPSO Version 4 Automated Aerosol Classification and Lidar Ratio Selection Algorithm" by Kim et al.

This manuscript details and evaluates changes to the new version 4 CALIOP aerosol classification algorithm. Major highlights, the inclusion of a "dusty marine" type and improvements to smoke classification in the troposphere, and a new stratospheric aerosol subtyping algorithm. Changes in the algorithm lead to improved extinction retrievals when compared to AERONET and MODIS over the ocean. The manuscript is well written and thoroughly details modifications to the algorithm. Therefore, I recommend that this manuscript be accepted following minor revision.

Major Comments:

[Figure]

1. For evaluating CALIOP extinction using MODIS, why not include MODIS deep blue AOD, which provides AOD over land, including bright surfaces? In particular, this might help to validate the impacts of improved surface detection in Figure 15e.

Minor Comments:

Page 2, Line 2: "Aerosol subtyping..." This sentence doesn't make sense to me.

Page 2, Lines 12 – 16: These findings (dust in marine boundary layer and smoke layers classified as marine) were also reported in Nowottnick et al., 2015. Consider adding a reference to this paper.

Page 5, Line 11: Is Rmas is defined as the ratio of the total attenuated backscatter to the molecular backscatter or the ratio of total attenuated backscatter to molecular attenuated backscatter? Some clarification would be useful to the reader. Also, is the molecular depolarization ratio assumed to be the same as in Omar et al., 2009? If so, please refer to that or provide to the reader.

Page 5, Lines 21-22: "First, ..." This sentence is confusing, please consider revising.

Page 6, Lines 9 – 12. In both V3 and V4 versions, why is polluted dust reduced over land compared to night? I'm assuming it's due to better signal-to-noise at night, but an explanation might be useful.

Page 6, Line 12: "AODdifferences". Missing a space.

Page 7, Line 9: I agree that a simple assumption of the PBL height is sufficient, but some justification for the threshold value (2.5 km) from literature or a figure that shows 2.5 km is a good global approximation should be included here.

Page 7, Lines 9-15: If I'm understanding the new modification to the algorithm correctly, polluted continental cannot be classified if the top of a layer is greater than 2.5 km. How does this impact the identification of long-range sulfate transport (ex. Asian pollution reaching North America)? This limitation might be noted here.

Page 8, Line 14: What is the integrated color ratio threshold for cloud vs. aerosol discrimination? The same as in Omar et al., 2009? Please include this in the discussion.

Page 9, Line 6: Consider explicitly stating the CALIPSO data and what version were used to make Figure 5.

Page 11, Lines 13-14. Is the tropopause altitude coming from GEOS output?

Page 11, Line 20. "aa", replace with "a".

Page 11, Lines 20-22. Does this misclassification have a significant impact on the extinction product?

Page 12, Line 4. "..", replace with "."

Page 14, Line 7. "releasesreleases", replace with "releases".

Page 16, Lines 6-9. Nice results using SCARF in the vertical. Is applying SCARF horizontally possible?

Page 19, Lines 23-24. Could you provide an example of detection of a tenuous layer?

Page 20, Line 4. "demonstratesan", replace with "demonstrates"

Page 22, Lines 19-20. There also is regions where polluted continental is the dominant aerosol type over southern Africa. So, the catch-all "PC/smoke" type works well in the region.

Page 23, Lines 11-12. Why does the overestimation of the estimated particulate depolarization ratio primarily affect dust/polluted dust over land?

Page 24, Line 18. "Whilewhile", replace with "While".

Page 24, Line 19. Consider omitting "correctly", since the occurrence of "dusty marine" cannot be independently validated.

Page 24, Line 24-25. What are there less nighttime stratospheric aerosol features than

during the day in V4?

Page 25, Line 8. "V3 or V4". Should this be "V3 and V4"?

Page 25, Line 22. What is "AVD"?

Page 28 – Why does the daytime AOD decrease from V3 to V4 over parts of the ocean, while the nighttime does not?

Page 30, Line 18. What is "STS"?

Page 32, Lines 21-23. "Resolution...CALIOP and MODIS over land are excluded from further consideration in our analyses." However, MODIS over land is compared to CALIOP in Table 8 and CALIOP AODs are biased higher in V4. Consider adding a sentence to explain this result. Also, see Major Comment #1 regarding the possible inclusion of MODIS deep blue in this evaluation.

Page 33, Line 8. Consider omitting "(perfect agreement....)."

Figure 16: What is causing the AOD hotspots in V4 over Alaska and central North America? The lidar ratios from V3 to V4 for elevated smoke are similar and improvements to the surface detection in this region show a decrease in AOD (Figure 15e).

Figures & Tables:

Figure 1: Caption – Consider forward referencing the justification for using 2.5 km as a threshold in the manuscript (page 6 in section 2.1.1 and section 2.1.3).

Table 1: Last line – "Canadians", replace with "Canadian".

Figure 7: Since depolarization and color ratio are inputs to the new stratospheric typing algorithm, consider adding those quantities to the figure.

Table 4: Consider adding "Tropo." to the "Aerosol" header in the 5th column.

Figure 15: Is it possible to adjust the color bar to resolve differences in V3 and V4 AOD more clearly (use a finer delta AOD)?

[Figure]

---

## Author Comment (AC4) · 26 Sep 2018

We thank the referee for their careful reading of our manuscript and their thoughtful comments. We have reproduced the referee's comments below (in black) and included our responses in-line (in blue).

Review of "The CALIPSO Version 4 Automated Aerosol Classification and Lidar Ratio Selection Algorithm" by Kim et al.

This manuscript details and evaluates changes to the new version 4 CALIOP aerosol classification algorithm. Major highlights, the inclusion of a "dusty marine" type and improvements to smoke classification in the troposphere, and a new stratospheric aerosol subtyping algorithm. Changes in the algorithm lead to improved extinction retrievals when compared to AERONET and MODIS over the ocean. The manuscript is well written and thoroughly details modifications to the algorithm. Therefore, I recommend that this manuscript be accepted following minor revision.

Major Comments:

1. For evaluating CALIOP extinction using MODIS, why not include MODIS deep blue AOD, which provides AOD over land, including bright surfaces?

The MODIS Deep Blue retrievals would undoubtedly be useful in evaluating CALIOP extinction and AOD estimates over desert regions and other highly reflective land surfaces. However, investigating regional variations in CALIOP AOD (or regional AOD differences between CALIOP and MODIS) is not an integral part of this study. Instead our primary goals are to (i) explain the mechanics of the V4 algorithm and highlight the ways that the V4 algorithm differs from the V3 algorithm; and (ii) illustrate the effects of the algorithm changes on the CALIOP data products. As part of doing part (ii), we use MODIS as a "constant external reference". And as long as we consistently use the same reference value (i.e., as long as we always compare V3 and V4 to the same MODIS data product), it really shouldn't matter which reference we choose.

Two factors motivate our choice of the MODIS Dark Target data set as our reference.

1. As noted in the discussion paper (see lines 24–25 on page 31), numerous previous publications have compared MODIS and CALIOP V3 AOD estimates. All of these studies used Dark Target; none used Deep Blue. By choosing Dark Target as our reference, we readily facilitate comparisons between our results and those published previously.

2. In discussing the reviewer's suggestion, we find ourselves highly persuaded by the Deep Blue vs. Dark Target comparison done by Sayer et al., 2014.

   *Due to the complexity of the global Earth (surface and atmospheric) system, and the optimization of the algorithms for global rather than regional applications, neither algorithm consistently performs better than the other. To make some general comments, over much of the global land surface, the AOD retrieved by the algorithms and the level of agreement with AERONET are similar, such that **it may not matter for many applications which of DB or***

*__DT a user chooses__. DT often has a better correlation with AERONET than DB, but DB has (outside of tropical regions) greater spatial coverage, and tends to have smaller error compared to AERONET values in low-AOD conditions. __DB and DT often exhibit much smaller AOD differences than would be expected given their estimated individual uncertainties__, which should not be taken to mean that they have converged on the truth, but is a reminder that they should not be considered to be independent data sets.*

*(The additional emphasis simply reinforces the point made earlier; i.e., 'as long as we consistently use the same reference value, it really shouldn't matter which reference we choose.')*

In particular, this might help to validate the impacts of improved surface detection in Figure 15e.

The "improved surface detection" in V4 allows CALIOP to retrieve estimates of aerosol extinction coefficients and optical depths in regions where the signal was previously considered to be totally attenuated by overlying layers that were mistakenly identified as being opaque. A huge majority of these overlying layers are clouds. Since MODIS cannot detect the aerosol layers below these clouds, MODIS data products cannot help evaluate the accuracy of CALIOP's subtype assignments in these aerosols.

Minor Comments:

Page 2, Line 2: "Aerosol subtyping..." This sentence doesn't make sense to me.

The original text was "Aerosol subtyping is important all by itself for identifying aerosol by type. But it is also important for the CALIOP level 2 retrievals of aerosol optical properties."

The revised text is "While CALIOP's aerosol subtype classifications are useful as a wholly independent data product (e.g., Nowottnick et al., 2015; Sun et al., 2018), knowledge of aerosol subtype is also critically important for the CALIOP level 2 retrievals of aerosol optical properties."

Page 2, Lines 12 – 16: These findings (dust in marine boundary layer and smoke layers classified as marine) were also reported in Nowottnick et al., 2015. Consider adding a reference to this paper.

Done

Page 5, Line 11: Is Rmas is defined as the ratio of the total attenuated backscatter to the molecular backscatter or the ratio of total attenuated backscatter to molecular attenuated backscatter? Some clarification would be useful to the reader. Also, is the molecular depolarization ratio assumed to be the same as in Omar et al., 2009? If so, please refer to that or provide to the reader.

Equation (1) in this manuscript is identical to equation (11) in Omar et al., 2009, as are the definitions of all quantities therein. As suggested, we have added a reference to the Omar paper.

Page 5, Lines 21-22: "First, . . ." This sentence is confusing, please consider revising.

The original text was "First, $\delta_p^{est}$ is a noisy quantity that has a positively skewed distribution that is exacerbated by solar background noise during the daytime."

The revised text is "First, $\delta_p^{est}$ is a noisy quantity that is asymmetrically distributed, with a large positively skewed tail that can be considerably increased by solar background noise during the daytime."

Page 6, Lines 9 – 12. In both V3 and V4 versions, why is polluted dust reduced over land compared to night? I'm assuming it's due to better signal-to-noise at night, but an explanation might be useful.

The dominant source of this difference is the day-night differences in depolarization ratio SNR. See our reply to the previous comment; larger daytime $\delta_p^{est}$ values result in more dust relative to polluted dust.

Page 6, Line 12: "AODdifferences". Missing a space.

Fixed

Page 7, Line 9: I agree that a simple assumption of the PBL height is sufficient, but some justification for the threshold value (2.5 km) from literature or a figure that shows 2.5 km is a good global approximation should be included here.

We added a reference to this paper: McGrath-Spangler, E. L., and A. S. Denning (2013), Global seasonal variations of midday planetary boundary layer depth from CALIPSO space-borne LIDAR, J. Geophys. Res. Atmos., 118, 1226–1233, doi:10.1002/jgrd.50198.

Page 7, Lines 9-15: If I'm understanding the new modification to the algorithm correctly, polluted continental cannot be classified if the top of a layer is greater than 2.5 km. How does this impact the identification of long-range sulfate transport (ex. Asian pollution reaching North America)? This limitation might be noted here.

The more general limitation is that pollution which is lofted above 2.5 km will be misclassified as smoke – or clean continental if the integrated attenuated backscatter is very low. To acknowledge this limitation, the following sentence was added to the end of section 2.1.3: "A limitation of identifying smoke layers according to altitude is that pollution lofted by convective processes or other vertical transport mechanisms can be misclassified as elevated smoke."

Page 8, Line 14: What is the integrated color ratio threshold for cloud vs. aerosol discrimination? The same as in Omar et al., 2009? Please include this in the discussion.

The color ratio threshold in the CALIOP cloud-aerosol discrimination (CAD) algorithm is not a single value, but is instead a function of latitude, longitude, and altitude. See Liu et al., 2009 and Liu et al., 2018 for details.

Page 9, Line 6: Consider explicitly stating the CALIPSO data and what version were used to make Figure 5.

We added "V4.1" to the modified caption for Figure 5. Note that figure 5 shows mid-layer temperatures for layers specifically identified as stratospheric aerosols (i.e., at temperatures below –70°C). As stated in the opening sentences of section 2.2, the classification of stratospheric layers as aerosols and clouds is a new feature first introduced in the version 4.1 data products. Previous releases did not report a stratospheric aerosol layer type.

Page 11, Lines 13-14. Is the tropopause altitude coming from GEOS output?

Yes

Page 11, Line 20. "aa", replace with "a".

Done

Page 11, Lines 20-22. Does this misclassification have a significant impact on the extinction product?

The lidar ratios currently specified for volcanic ash and dust are identical at 44 sr for both wavelengths, so in this situation the extinction coefficient and optical depth estimates will be identical irrespective of the aerosol type assigned.

Page 12, Line 4. "..", replace with "."

Done

Page 14, Line 7. "releasesreleases", replace with "releases".

Done

Page 16, Lines 6-9. Nice results using SCARF in the vertical. Is applying SCARF horizontally possible?

Perhaps…but that's something we wouldn't consider attempting until the version 5 data release at the very earliest

Page 19, Lines 23-24. Could you provide an example of detection of a tenuous layer?

See Figure 13 in Liu et al., 2018

Page 20, Line 4. "demonstratesan", replace with "demonstrates"

Done

Page 22, Lines 19-20. There also is regions where polluted continental is the dominant aerosol type over southern Africa. So, the catch-all "PC/smoke" type works well in the region.

Page 23, Lines 11-12. Why does the overestimation of the estimated particulate depolarization ratio primarily affect dust/polluted dust over land?

This happens because the relative fraction of dust and polluted dust detected over land is typically much higher than over water.

Page 24, Line 18. "Whilewhile", replace with "While".

Done

Page 24, Line 19. Consider omitting "correctly", since the occurrence of "dusty marine" cannot be independently validated.

We replaced "correctly" with "more realistically"

Page 24, Line 24-25. What are there less nighttime stratospheric aerosol features than during the day in V4?

To reduce/eliminate potential confusion, we have rewritten lines 23–25. The original sentences read as follows: "Stratospheric features detected in V3 are classified as cloud or stratospheric aerosol in V4. Stratospheric features in V3 are changed to cloud more frequently for nighttime. Similarly, stratospheric aerosol in V4 is less at nighttime than daytime compared to stratospheric features in V3." This is the new text: "The generic stratospheric features previously identified in V3 are now classified as clouds or aerosol in V4. During the daytime, these V3 stratospheric features are more frequently identified as clouds, rather than aerosols. At night the situation is reversed: nighttime V3 stratospheric features are most often classified as aerosols."

Page 25, Line 8. "V3 or V4". Should this be "V3 and V4"?

The original text was "either/both V3 or V4". The revised text is "either V3 or V4".

Page 25, Line 22. What is "AVD"?

Atmospheric volume description; see page 20, line 12 in the discussion paper

Page 28 – Why does the daytime AOD decrease from V3 to V4 over parts of the ocean, while the nighttime does not?

In V3, mixtures of dust and marine aerosol were misclassified as polluted dust more frequently in the daytime relative to night. This is because of the noisiness of the daytime signal. These dust/marine mixtures were reclassified as dusty marine in V4, with a smaller lidar ratio. Since more layers changed from polluted dust to dusty marine in the daytime relative to night, the reduction in AOD is greater in the day over some parts of the ocean. During the night, the change in AOD is dominated by the increase in marine lidar ratio.

Page 30, Line 18. What is "STS"?

Supercooled ternary solution; see page 8, line 8 in the discussion paper

Page 32, Lines 21-23. "Resolution. . .CALIOP and MODIS over land are excluded from further consideration in our analyses." However, MODIS over land is compared to CALIOP in Table 8 and CALIOP AODs are biased higher in V4. Consider adding a sentence to explain this result. Also, see Major Comment #1 regarding the possible inclusion of MODIS deep blue in this evaluation.

We made some initial comparisons of CALIOP, MODIS, and AERONET AODs retrieved over land. These comparisons are illustrated in Figure 16 and summarized in Table 8. As we state in the paper, "the inconsistency between the comparisons with AERONET and MODIS points to the need for further validation studies, especially over land." This is the justification provided for limiting additional CALIOP–MODIS comparisons to over oceans only, where MODIS retrievals are considerably more reliable. We revisited our reasoning in light of your comments, and remain convinced that our initial choice was (and is) appropriate.

Page 33, Line 8. Consider omitting "(perfect agreement. . ..)."

The phrase is correct as written; "perfect agreement" would indeed result in a slope of zero.

Figure 16: What is causing the AOD hotspots in V4 over Alaska and central North America? The lidar ratios from V3 to V4 for elevated smoke are similar and improvements to the surface detection in this region show a decrease in AOD (Figure 15e).

Figure 15e only shows the changes in AOD that can be attributed to improved surface detection (i.e., as determined according to the flowchart in Figure 14). With regard to Figure 16, the more relevant panel in Figure 15 is the daytime (right-hand) panel of 15a, which shows the total CALIOP AOD change from V3 to V4. (The nighttime panel of 15a is irrelevant in this context, because Figure 16 shows comparisons to MODIS, which only estimated AOD during the daytime.) According to the color scale in Figure 15, the increases in CALIOP daytime AOD in the "hot spot" regions are somewhere between 0.05 and 0.10. This seems to us to be consistent with the changes shown in Figure 16; e.g., the V3 AODs in central North America range between ~0.050 lower than MODIS (pale blue color) to 0.125 higher (reddish brown color), whereas the V4 AODs are between ~0.05 and ~0.15 higher.

Figures & Tables:

Figure 1: Caption – Consider forward referencing the justification for using 2.5 km as a threshold in the manuscript (page 6 in section 2.1.1 and section 2.1.3).

We added a reference to section 2.1.3

Table 1: Last line – "Canadians", replace with "Canadian".

We changed "Canadians fire" to "Canadian fires"

Figure 7: Since depolarization and color ratio are inputs to the new stratospheric typing algorithm, consider adding those quantities to the figure.

It's our opinion that Figures 7, 8, and 9 provide sufficient information to illustrate our points, and would be made overly-complicated by adding even more panels. Instead we have added text to the captions of these figures saying that "Additional imagery for this scene, including 532 nm depolarization ratios and attenuated backscatter color ratios, can be found at https://www-calipso.larc.nasa.gov/..."

Table 4: Consider adding "Tropo." to the "Aerosol" header in the 5th column.

Since V3 did not separately identify stratospheric aerosols, this additional notation would be redundant.

Figure 15: Is it possible to adjust the color bar to resolve differences in V3 and V4 AOD more clearly (use a finer delta AOD)?

We experimented with **LOTS** of different color bars for this figure, and the one we've used, while still not perfect, was nevertheless the best of the bunch.
* * *
**References**

Liu, Z., Vaughan, M., Winker, D., Kittaka, C., Getzewich, B., Kuehn, R., Omar, A., Powell, K., Trepte, C and Hostetler, C.: The CALIPSO lidar cloud and aerosol discrimination: Version 2 algorithm and initial assessment of performance, J. Atmos. Oceanic Technol., 26, 1198–1213, doi:10.1175/2009JTECHA1229.1, 2009.

Liu, Z., Kar, J., Zeng, S., Tackett, J., Vaughan, M., Avery, M., Pelon, J., Getzewich, B., Lee, K.-P., Magill, B., Omar, A., Lucker, P., Trepte, C., and Winker, D.: Discriminating Between Clouds and Aerosols in the CALIOP Version 4.1 Data Products, Atmos. Meas. Tech. Discuss., doi:10.5194/amt-2018-190, in review, 2018.

Nowottnick, E. P., Colarco, P. R., Welton, E. J., and Da Silva, A.: Use of the CALIOP vertical feature mask for evaluating global aerosol models, Atmos. Meas. Tech., 8(9), 3647-3669, doi:10.5194/amt-8-3647-2015, 2015.

Sayer, A. M., Munchak, L. A., Hsu, N. C., Levy, R. C., Bettenhausen, C., and Jeong, M.-J.: MODIS Collection 6 aerosol products: Comparison between Aqua's e-Deep Blue, Dark Target, and "merged" data sets, and usage recommendations, J. Geophys. Res. Atmos., 119, 13,965–13,989, doi:10.1002/2014JD022453, 2014.

Sun, T., Che, H., Qi, B., Wang, Y., Dong, Y., Xia, X., Wang, H., Gui, K., Zheng, Y., Zhao, H., Ma, Q., Du, R., and Zhang, X.: Aerosol optical characteristics and their vertical distributions under enhanced haze pollution events: effect of the regional transport of different aerosol types over eastern China, Atmos. Chem. Phys., 18, 2949–2971, doi:10.5194/acp-18-2949-2018, 2018.